EMBO
Molecular Medicine

# *ALX1*-related frontonasal dysplasia results from defective neural crest cell development and migration

Jonathan Pini[1,2,†], Janina Kueper[1,2,3,†], Yiyuan David Hu[1,2], Kenta Kawasaki[1,2], Pan Yeung[1,2], Casey Tsimbal[1,2], Baul Yoon[4], Nikkola Carmichael[5], Richard L Maas[5], Justin Cotney[6], Yevgenya Grinblat[4] & Eric C Liao[1,2,*]

## Abstract

A pedigree of subjects presented with frontonasal dysplasia (FND). Genome sequencing and analysis identified a p.L165F missense variant in the homeodomain of the transcription factor *ALX1* which was imputed to be pathogenic. Induced pluripotent stem cells (iPSC) were derived from the subjects and differentiated to neural crest cells (NCC). NCC derived from ALX1[L165F/L165F] iPSC were more sensitive to apoptosis, showed an elevated expression of several neural crest progenitor state markers, and exhibited impaired migration compared to wild-type controls. NCC migration was evaluated *in vivo* using lineage tracing in a zebrafish model, which revealed defective migration of the anterior NCC stream that contributes to the median portion of the anterior neurocranium, phenocopying the clinical presentation. Analysis of human NCC culture media revealed a change in the level of bone morphogenic proteins (BMP), with a low level of BMP2 and a high level of BMP9. Soluble BMP2 and BMP9 antagonist treatments were able to rescue the defective migration phenotype. Taken together, these results demonstrate a mechanistic requirement of *ALX1* in NCC development and migration.

**Keywords** ALX1; frontonasal dysplasia; iPSC; neural crest cells; zebrafish
**Subject Categories** Development; Genetics, Gene Therapy & Genetic Disease; Stem Cells & Regenerative Medicine

## Introduction

The central part of the human face contains key anatomic features and sensory organs that enable us to interact with the environment and each other. The embryologic processes that form midface structures, including the eyes, nose, upper lip, and maxilla, are tightly regulated (Johnston, 1966; Minoux & Rijli, 2010; Rada-Iglesias *et al*, 2012). The midface structures form as the centrally located frontonasal prominence extends anteriorly, coalescing with elements derived from the paired maxillary prominences (Johnston, 1966, 1975; Le Lièvre & Le Douarin, 1975; Le Lièvre, 1978; Sadaghiani & Thiébaud, 1987). The embryonic facial prominences are derived from distinct migrating streams of cranial neural crest cells (NCC) that are conserved across vertebrates (Le Douarin *et al*, 1993; Schilling *et al*, 1996; Chai *et al*, 2000; Olsson, *et al*, 2002; Trainor *et al*, 2002; Barrallo-Gimeno *et al*, 2004; Wada, 2005; Dougherty *et al*, 2012). NCC migration and differentiation are highly coordinated and are associated with dynamic gene expression patterns (Simoes-Costa & Bronner, 2015). Key signaling pathways that regulate NCC development involve BMP, Wnt, FGF, or Notch which activate the expression of transcription factors such as *PAX3*, *ZIC1*, *TFAP2a*, *MSX1/2,* and *DLX5* (Meulemans & Bronner-Fraser, 2002; Khudyakov & Bronner-Fraser, 2009; Stuhlmiller & Garcia-Castro, 2012; Rada-Iglesias *et al*, 2013; Simoes-Costa & Bronner, 2015). Disruptions of NCC development contribute to a number of congenital malformations such as Waardenburg syndrome (WS), velo-cardiofacial syndrome/DiGeorge syndrome, Hirschsprung's disease, congenital heart conditions, and craniofacial anomalies (Sedano *et al*, 1970; Fox *et al*, 1976; Pierpont *et al*, 2007; Uz *et al*, 2010)

Frontonasal dysplasia (FND) is considered a rare "orphan" disease (ORPHA250), with very few cases reported in the literature. The true prevalence of FND and its etiology remain unknown. To date, six genetic causes of subtypes of FND with varying patterns of

1 Center for Regenerative Medicine, Department of Surgery, Massachusetts General Hospital, Boston, MA, USA
2 Shriners Hospital for Children, Boston, MA, USA
3 Life and Brain Center, University of Bonn, Bonn, Germany
4 Departments of Integrative Biology, Neuroscience, and Genetics Ph.D. Training Program, University of Wisconsin-Madison, Madison, WI, USA
5 Department of Genetics, Brigham and Women's Hospital, Harvard Medical School, Boston, MA, USA
6 Genetics and Genome Sciences, UConn Health, Farmington, CT, USA
*Corresponding author. Tel: +1 6176435975; E-mail: cliao@partners.org
†These authors contributed equally to this work

inheritance have been described in individual case reports: *EFNB1* (MIM 300035) in X-linked craniofrontonasal syndrome (MIM 304110); *ALX3* (MIM 606014) in FND type 1 (MIM 136760); *ALX4* (MIM 605420) in FND type 2 (MIM 613451); *ALX1* (MIM 601527) in FND type 3 (MIM 613456); *ZSWIM6* (MIM 615951) in dominant acromelic frontonasal dysostosis (MIM 603671); and *SPECC1L* (MIM 614140) in Teebi syndrome (MIM 145420; Bhoj *et al*, 2015; Kayserili *et al*, 2009; Smith *et al*, 2014; Twigg *et al*, 2004, 2009; Ullah *et al*, 2016; Uz *et al*, 2010; Wieland *et al*, 2004). The heterogeneity of clinical phenotypes, including a wide range of possible ocular and craniofacial components, likely corresponds to different underlying genetic variants, genetic environments, and epigenetic modifications.

This study examined a pedigree of FND and identified a pathogenic variant in the homeodomain of transcription factor ALX1 resulting in a likely loss of function. A human stem cell model of FND was generated in order to investigate the effect of *ALX1* mutations on NCC behavior. Cellular and molecular characterizations identified a number of differences between subject-derived and control NCC that shed light on the developmental processes that are disrupted in FND. *In vivo* characterization of *alx1* in zebrafish revealed defective migration of the most anterior cranial NCC. This study underscores the utility of complementary human iPSC and zebrafish models to gain mechanistic insight into the molecular and cellular basis of *ALX1*-related FND.

# Results

## Clinical features of ALX1-related FND in a consanguineous pedigree

A family with four children born of consanguineous parents of Amish heritage presented with complex FND. The FND phenotype was inherited in a Mendelian recessive fashion. Both parents, one unaffected sibling and three affected children (one male and two females), were consented and enrolled in the study (Fig 1A, subject numbers indicated in red). The parents (subjects 1 and 2) and nine unaffected siblings (including subject 3) had normal facial structures without clinical stigmata suggestive of mild FND. All four affected children presented with bilateral oblique facial clefts, extending from either side of the nasal bone, involving both the primary and secondary palate. Among the affected children, there was some variability of the ocular phenotype, where the older affected girl (subject 4) presented with bilateral coloboma and asymmetric microphthalmia, whereas the three other affected children (including subjects 5 and 6) exhibited bilateral anophthalmia, with deficient upper and lower eyelids covering a shallow orbit. Subject 6 was the most severely affected and presented with bilateral oblique facial clefts and anophthalmia as well as no upper and lower eyelids, leaving the mucous membranes of both orbits exposed. Her nasal remnant also lacked the lateral alar subunits and is surrounded by several nodular skin tags.

## Identification of pathogenic *ALX1* variant

Whole-exome sequencing (WES) was performed on blood samples collected from subjects 1–5, which corresponded to both parents, one unaffected sibling, and two affected children. A missense p.L165F variant (c.493C>T) was identified in the homeodomain of *ALX1*, which was heterozygous in the parents (ALX1$^{165L/165F}$), wild type in the unaffected sibling (ALX1$^{165L/165L}$), and homozygous in both affected subjects (ALX1$^{165F/165F}$; Fig 1B). WES results were confirmed by Sanger sequencing of the entire *ALX1* coding sequence. The ALX1 p.L165F missense variant has not been reported in connection with an *ALX1*-related instance of FND in the literature nor been recorded as a variant in the gnomAD database

---

**Figure 1. Clinical presentation of the FND pedigree and generation of control, father, and subject-derived iPSC.**

A The pedigree family tree includes two unaffected parents, four unaffected male siblings, five unaffected female siblings, and two each female and male affected sibling. Subjects 1–6, indicated in red, were enrolled in the study. Subjects 4–6 show complex FND with ocular involvement. The eldest affected sibling (subject 4) presented with right coloboma, left microphthalmia, and bilateral Tessier 4 oblique facial clefts. Subject 5 presented with bilateral anophthalmia with fused eyelids and shallow orbits, with bilateral oblique facial clefts. Subject 6 presented with bilateral anophthalmia with open shallow orbits, absent upper and lower eyelids, exposed orbital mucosa, bilateral oblique facial clefts, and malformed nasal ala with nodular skin tags. iPSCs were generated using blood samples collected from subjects 1, 5, and 6.

B Whole-exome sequencing was carried out and analysis revealed a missense p.L165F variant (c.493 C>T) in the *ALX1* homeodomain, heterozygous in the parents (*ALX1*$^{165L/165F}$), wild type in the unaffected sibling (*ALX1*$^{165L/165L}$), and homozygous in both affected subjects (*ALX1*$^{165F/165F}$).

C Schematic of the ALX1 protein structure showing the position of the L165F substitution described here (red) and the locations of exon borders affected by two reported pathogenic variants (purple; Ullah *et al*, 2016; Uz *et al*, 2010).

D Schematic of the *ALX1* genomic sequence, showing the locations of the three reported pathogenic variants. The purple bar at the bottom represents a FND-associated homozygous *ALX1* deletion previously reported in the literature (Uz *et al*, 2010; Ullah *et al*, 2016).

E Immunofluorescence staining for pluripotent markers *SSEA4*, *OCT4*, *SOX2*, and *TRA-1-60* and alkaline phosphatase staining of iPSC clones. One representative iPSC clone is shown for each genotype. Scale bar: 400 μm.

F Expression of pluripotent (OCT4, NANOG), endoderm (Endo., AFP, GATA4, FOXA2), ectoderm (Ecto., NESTIN, GFAP, SOX1), and mesoderm (Meso., BRACH. (BRACHYURY), RUNX1, CD34) gene markers for *ALX1*$^{165L/165L}$ (green), *ALX1*$^{165L/165F}$ (red), and *ALX1*$^{165F/165F}$ (blue) iPSC relative to undifferentiated cells (UND). Data are represented as pooled mean $\pm$ SEM of three experiments on three clones from each genotype. Significance: $P = 0.0167$ for OCT4, $P = 0.0005$ for NANOG, $P = 0.000004$ for AFP, $P = 0.0082$ for GATA4, $P = 0.0137$ for FOXA2, $P = 0.00002$ for NESTIN, $P = 0.0167$ for GFAP, $P = 0.0014$ for SOX1, $P = 0.0117$ for BRACHYURY, $P = 0.0008$ for RUNX1 and $P = 0.0068$ for CD34 when comparing undifferentiated and differentiated *ALX1*$^{165L/165L}$ iPSC. $P = 0.0013$ for OCT4, $P = 0.0011$ for NANOG, $P = 0.0000003$ for AFP, $P = 0.0003$ for GATA4, $P = 0.0063$ for FOXA2, $P = 0.0001$ for NESTIN, $P = 0.027$ for GFAP, $P = 0.000002$ for SOX1, $P = 0.000009$ for BRACHYURY, $P = 3e^{-9}$ for RUNX1 and $P = 0.000006$ for CD34 when comparing undifferentiated and differentiated ALX1$^{165F/165L}$ iPSC. $P = 0.0201$ for OCT4, $P = 0.006$ for NANOG, $P = 1 \times 10^{-12}$ for AFP, $P = 5 \times 10^{-13}$ for GATA4, $P = 0.0031$ for FOXA2, $P = 0.0292$ for NESTIN, $P = 0.00001$ for GFAP, $P = 6 \times 10^{-7}$ for SOX1, $P = 0.0204$ for BRACHYURY, $P = 0.0009$ for RUNX1 and $P = 0.000003$ for CD34 when comparing undifferentiated and differentiated *ALX1*$^{165F/165F}$ iPSC. Data from each clone were pooled, and the mathematical mean was calculated. SEM was used to determine the standard error. To test statistical significance, an ANOVA test was performed. A *P*-value $< 0.05$ was considered to be statistically significant.

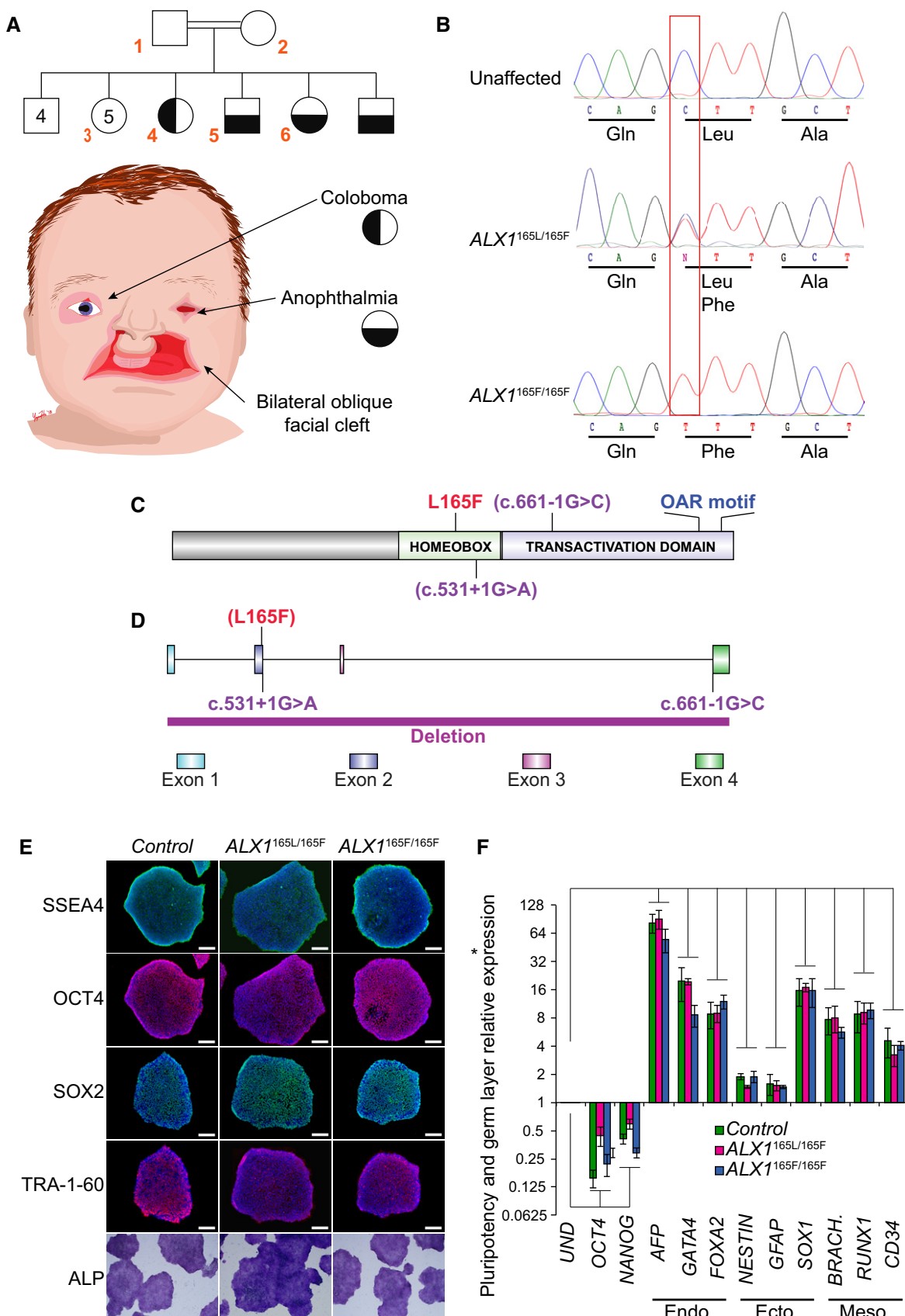

**Figure 1.**

(preprint: Karczewski et al, 2019; Fig 1C and D). The ALX1 p.L165F amino acid substitution was predicted to be damaging and disease causing by in silico tools (Sift, Polyphen, muttaster, fathmm) and consistent with the observed autosomal recessive inheritance pattern of this pedigree (Lowe, 1999; Adzhubei et al, 2010; Schwarz et al, 2014; Shihab et al, 2014).

### Generation of patient-derived iPSC model of ALX1-related FND

Induced pluripotent stem cell lines were generated using peripheral blood mononuclear cells (PBMC) obtained from whole blood samples that were collected from three unrelated wild-type individuals ($ALX1^{165L/165L}$), the heterozygous father ($ALX1^{165L/165F}$), and two of the four affected children (Subjects 5 and 6; $ALX1^{165F/165F}$). PBMC were subsequently reprogrammed into iPSC (Fig EV1). Overall, 22 mutant $ALX1^{165F/165F}$ iPSC clones were successfully isolated and expanded from the two affected subjects, 13 $ALX1^{165L/165F}$ clones were isolated and expanded from the heterozygous father, and 35 $ALX1^{165L/165L}$ clones were isolated and expanded from healthy controls. Six $ALX1^{165F/165F}$ mutant clones (3 for each affected subject), 3 $ALX1^{165L/165F}$ clones from the heterozygous father, and 9 $ALX1^{165L/165L}$ clones from healthy controls (3 from each control) were fully characterized to confirm their pluripotency (Fig 1E) and ability to generate the three germ layers (Fig 1F). Sanger sequencing confirmed that the affected $ALX1^{165F/165F}$ and the heterozygous $ALX1^{165L/165F}$-derived iPSC clones retained the ALX1 p.L165F variant through reprogramming. Copy number variant analysis did not show any amplifications or deletions.

### Generation and characterization of iPSC-derived NCC

Given the primary role of neural crest cells in midface morphogenesis, the iPSC clones were differentiated into NCC using a protocol adapted from a previous study (Pini et al, 2018; Fig 2A). All NCC displayed similar morphological features and were indistinguishable at the colony level immediately following differentiation at passage 1 (Fig 2B).

### Overexpression of neural plate border specifier genes in $ALX1^{165F/165F}$ NCC

A panel of marker genes at the center of the gene regulatory network required for NCC development and differentiation was selected to be examined in detail across the 14 days of the neural crest differentiation protocol (Fig 3; Barrallo-Gimeno et al, 2004; Sauka-Spengler & Bronner-Fraser, 2008; Sauka-Spengler et al, 2007; Simoes-Costa & Bronner, 2015). The NCC gene expression results can broadly be divided into three groups. The first group includes genes that did not significantly differ between affected, heterozygous, and unaffected controls. This group of genes comprises the

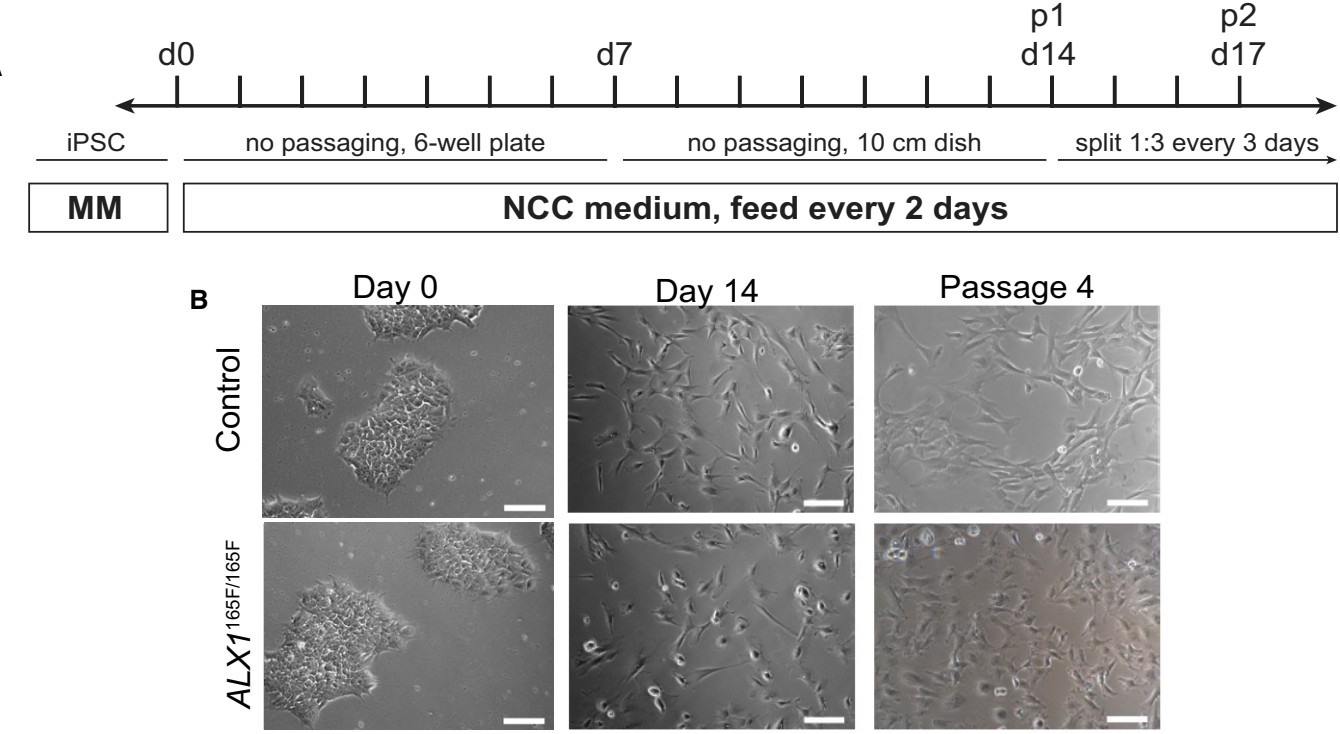

**Figure 2. Generation of iPSC-derived NCC.**

A Schematic of the differentiation protocol timeline. Maintenance Medium (MM) = iPSC medium (StemFlex with 1× penicillin/streptomycin), NCC differentiation medium = DMEM-F12, 10% fetal bovine serum, 1 mM sodium pyruvate, 1 mM penicillin/streptomycin, 1 mM nonessential amino acids, 110 μM 2-mercaptoethanol, 10 ng/ml epidermal growth factor.

B Images of iPSC and iPSC-derived NCC at Days 0, 14, and passage 4 following differentiation. Scale bars: 400 μm (Day 0), 200 μm (Day 14, passage 4).

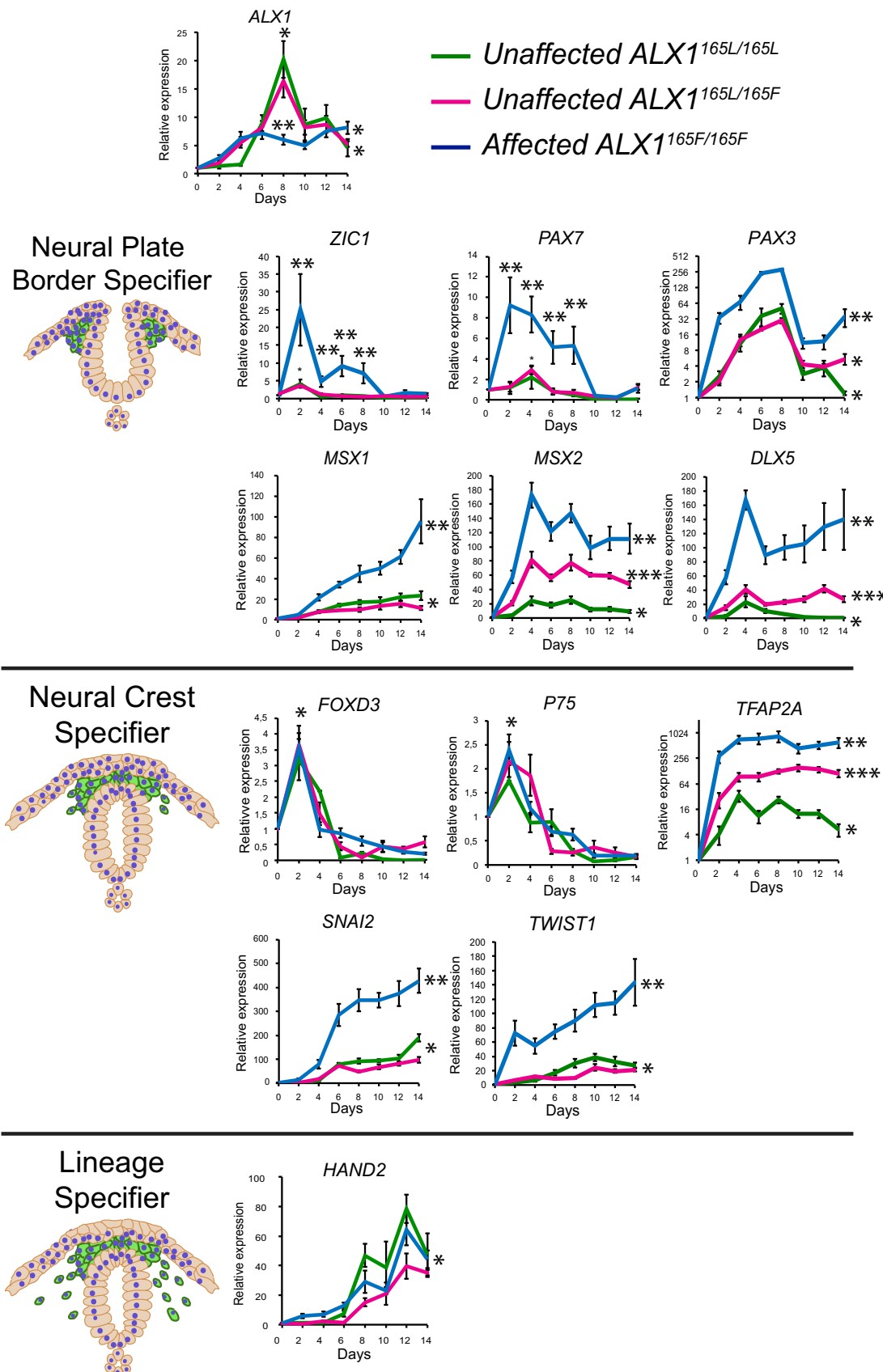

**Figure 3.**

◄

**Figure 3. Timeline of key NCC-associated genes during differentiation.**

Gene expression analysis across NCC differentiation of unaffected control $ALX1^{165L/165L}$ (green), heterozygous $ALX1^{165L/165F}$ (magenta), and homozygous $ALX1^{165F/165F}$ iPSC: *ALX1*, neural plate border specifier genes *ZIC1, PAX7, PAX3, MSX1, MSX2, DLX5*; neural crest specifier genes *FOXD3, P75, TFAP2A, SNAI2, TWIST1*; and lineage specifier gene *HAND2*. The RT–qPCR relative expression values were normalized to *RPLP0* and *GAPDH* expression. Data are represented as pooled mean ± SEM of three experiments on three clones from each genotype. Exact *P*-values are provided in Table EV1. Data from each clone were pooled, and the mathematical mean was calculated. SEM was used to determine the standard error. To test statistical significance, an ANOVA test was performed. A *P*-value < 0.05 was considered to be statistically significant.

neural crest specifiers *FOXD3* and *P75*, as well as the lineage specifier *HAND2*. The second group includes genes where the affected cells exhibited expression patterns that differed significantly from the heterozygous and the unaffected control cells, with no difference between the heterozygote and the control. This group of genes includes the neural plate border specifiers *ZIC1, PAX7, PAX3, MSX1*, and *DLX5* and the neural crest specifiers *SNAI2* and *TWIST1* ($P < 0.05$ between days 2–8 when comparing subjects', father, and control NCC for *ZIC1, PAX7, DLX5*; $P < 0.05$ between days 2–14 when comparing subjects', father, and control NCC for *PAX3, MSX1, SNAI2*, and *TWIST1*). The final group includes genes that were significantly differentially expressed between the affected homozygous, heterozygous, and unaffected control cells. This group comprised the neural plate border specifier *MSX2, DLX5*, and the neural crest specifier *TFAP2A* ($P < 0.05$ between days 2–14 when comparing subjects', father, and control NCC for *MSX2, DLX5*, and *TFAP2A*). Of note, all significantly differentially expressed genes in the affected cells were overexpressed above the levels observed in the heterozygous and unaffected control cells, consistent with a putative role of ALX1 as a transcriptional repressor.

*ALX1* itself was found to be differentially expressed between affected cells when compared to the heterozygous and unaffected control cells at day 8 during NCC differentiation. The unaffected control and heterozygous cells exhibited similar *ALX1* expression levels, with peak expression level reached at day 8 where unaffected cells exhibited a plateaued, lower level of expression. The greatest difference in gene expression levels was observed early in the NCC differentiation process, around days 2–8 (such as in the cases of *ZIC1, PAX3, PAX7, DLX5*, and *TWIST1*). This characterization

suggests an early function for ALX1 in NCC differentiation and identifies the 2–8 day window for in-depth transcriptome analysis in future studies.

## Increased sensitivity to apoptosis in ALX1$^{165F/165F}$ NCC

Since anomalies in cell cycle progression predispose cells to apoptosis and given the importance of apoptosis in regulating craniofacial development, the impact of the $ALX1^{165F/165F}$ gene variant on the sensitivity of the iPSC-derived NCC to apoptosis was analyzed. Basal apoptosis levels, determined by the percentage of Annexin V-positive cells, did not differ between control NCC (4 ± 0.2%) and $ALX1^{165F/165F}$ NCC (4.82 ± 0.65%; Fig 4A). After apoptosis induction via heat shock, the percentage of Annexin V-positive cells significantly increased specifically in $ALX1^{165F/165F}$ NCC (87.97 ± 2.44%) versus control NCC (24.15 ± 0.96%). These findings suggest that the affected subject's $ALX1^{165F/165F}$ NCC are more sensitive to apoptosis.

These findings also indicate that *ALX1* functions in proliferating NCC. To determine whether *ALX1* function is required for cell cycle progression, we investigated expression of Cyclin D1 (CCND1), required for the cell cycle G1/S transition, and Cyclin A2 (CCNA2), required for the DNA synthesis during the S-phase. Both cyclins are expressed throughout the active cell cycle, from the G1/S transition to the G2/M transition (Pagano *et al*, 1992; Minarikova *et al*, 2016). Expression levels of CCND1 and CCNA2 were compared between control and ALX1$^{165F/165F}$ NCC at passages 2 and 3 post-differentiation. The $ALX1^{165F/165F}$ NCC were found to express significantly more CCNA2 and CCND1 at both passage 2 and passage 3 compared

**Figure 4. NCC apoptosis, cell cycle, and differentiation.**

A Homozygous $ALX1^{165F/165F}$ NCC (blue) showed an increase in sensitivity to apoptosis when compared to control $ALX1^{165L/165L}$ NCC (black). The data on the left represent the mean percentage of Annexin V-positive cells, indicative of apoptosis, as determined by FACS analysis, with the data on the right being an example of one such experiment. Apoptosis was induced by immersion in a 55°C water bath for 10 min. Representative experiment for each condition is shown. Data are represented as pooled mean ± SEM of three independent experiments. Data obtained of each clone from three independent experiments were pooled, and the mathematical mean was calculated. SEM was used to determine the standard error. To test statistical significance, an ANOVA test was performed. A *P*-value < 0.05 was considered to be statistically significant. *: Significantly different from the basal apoptosis rate: $P = 3 \times 10^{-12}$ between control NCC basal apoptosis and induced apoptosis, and between $ALX1^{165F/165F}$ NCC basal apoptosis and induced apoptosis. **Significantly different from control NCC ($P = 0.0004$).

B Expression levels of cyclins *CCNA2* (blue) and *CCND1* (orange) in NCC at passages 2 and 3 of $ALX1^{165L/165L}$ and $ALX1^{165F/165F}$ NCC. The RT–qPCR relative expression values were normalized to *RPLP0* and *GAPDH* expression. Data are represented as pooled mean ± SEM of three experiments on three clones from each genotype. *Significantly different from control NCC at passage 2 ($P = 0.001$ between control and $ALX1^{165F/165F}$ NCC at passage 2 for CCNA2, $P = 0.0052$ for CCND1). **Significantly different from control NCC at passage 3 ($P = 0.0494$ between control and $ALX1^{165F/165F}$ NCC for CCNA2, $P = 0.0008$ for CCND1).

C Fluorescence activated cell sorting (FACS) experiments showed that control $ALX1^{165L/165L}$ NCC (green) exhibited increased expression of mesenchymal markers CD90, CD105, and CD73 with culture time (passages 1 through 4), whereas homozygous $ALX1^{165F/165F}$ NCC (blue) showed a consistent expression of the markers expressed at passage 1 throughout. Further, control $ALX1^{165L/165L}$ NCC showed a downregulation of CD57 expression with culture time, while $ALX1^{165F/165F}$ NCC maintained the same level of CD57 across passages. Data are presented as the mean percentage of positive-stained cells across passage numbers. Data obtained of each clone from three independent experiments were pooled, and the mathematical mean was calculated. SEM was used to determine the standard error. To test statistical significance, an ANOVA test was performed. A *P*-value < 0.05 was considered to be statistically significant. *Significantly different from control NCC. For CD90, $P = 0.0013$ at passage 3 and $P = 0.0207$ at passage 4. For CD105, $P = 0.0016$ at passage 2, $P = 0.00004$ at passage 3 and $P = 0.0021$ at passage 4. For CD73, $P = 0.0060$ at passage 2, $P = 0.00004$ at passage 3 and $P = 0.0114$ at passage 4. For CD57, $P = 0.0026$ at passage 2, $P = 0.000003$ at passage 3 and $P = 0.000007$ at passage 4.

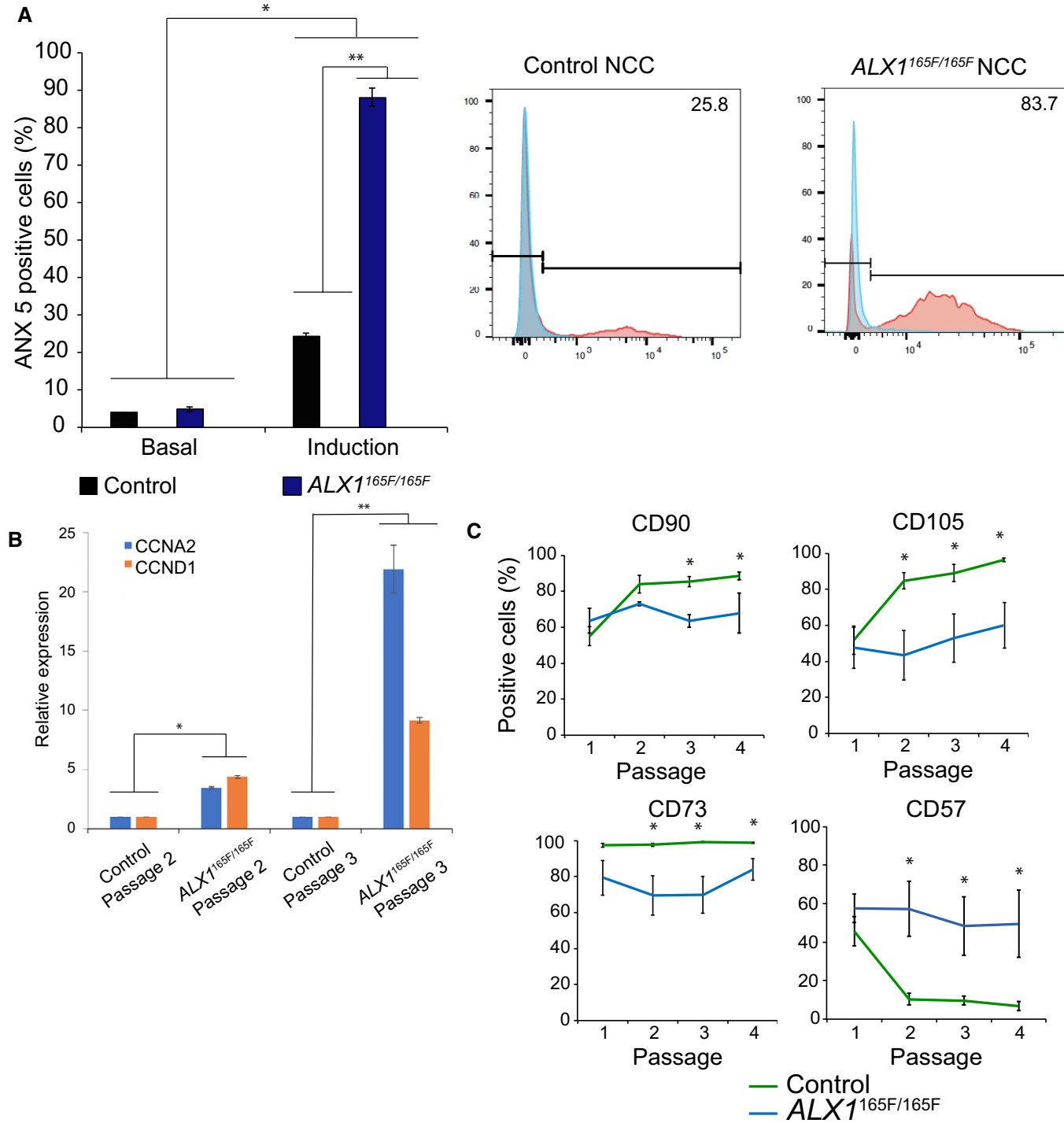

**Figure 4.**

to the control NCC, consistent with a greater degree of active cellular proliferation (Fig 4B).

### ALX1[165F/165F] NCC do not undergo mesenchymal marker transition

As NCC clones derived from the control *ALX1*[165L/165L], heterozygous *ALX1*[165L/165F], and homozygous *ALX1*[165F/165F] iPSC were

maintained in culture, consistent qualitative morphologic differences were observed across cell passages. While control-derived NCC became progressively spindle-shaped and elongated, the mutant *ALX1*[165F/165F] NCC remained rounded (Fig 2B). In order to investigate these differences more fully, flow cytometry was performed across different cell passage cycles in order to investigate the effect of the *in vitro* maturation of the NCC via an examination of NCC marker expression. At passage cycles 1–4, a number of key

surface markers were examined. Expression of CD57 (synonym: HNK-1), indicative of NCC precursors before their commitment to downstream cell lineages (Minarcik & Golden, 2003), as well as markers of mesenchymal differentiation, CD105, CD73, and CD90, was assessed (Fig 4C).

Table 1 contains the precise percentage values of the FACS analysis of NCC at varying passage numbers. At passage 1 following differentiation, control and homozygous $ALX1^{165F/165F}$ NCC expressed similar levels of neural crest precursor marker CD57. The control and homozygous $ALX1^{165F/165F}$ NCC also expressed similar levels of mesenchymal markers CD90, CD105, and CD73. No significant differences were observed in marker expression between control and homozygous $ALX1^{165F/165F}$ NCC at this stage ($P > 0.05$). However, by passage 4, control NCC exhibited decreased CD57 expression and increased expression of CD90, CD105, and CD73, consistent with a progression to MSC differentiation. In contrast, homozygous $ALX1^{165F/165F}$ NCC displayed a similar expression of the aforementioned NCC and MSC markers at passage 4 as they did at passage 1 (Fig 4C).

The persistent CD57 expression in the homozygous $ALX1^{165F/165F}$ NCC, taken together with the elevated expression of neural crest specifier genes *ZIC1, PAX7, PAX3, MSX1, MSX2, and DLX5,* suggests that the mutant NCC may be unable to progress from the progenitor to the differentiating state. To understand whether persistent CD57 expression had an effect on the ability of the homozygous $ALX1^{165F/165F}$ NCC to differentiate into downstream cell types, multilineage differentiation experiment was performed. Control and homozygous $ALX1^{165F/165F}$ NCC demonstrated equal ability to differentiate into Schwann cells (GFAP and S100B-positive expression), adipocytes (oil Red O. staining), chondrocytes (Alcian Blue, Safranin O. and Toluidine Blue staining), and osteoblasts (Alizarin Red S., Von Kossa staining and strong alkaline phosphatase activity; Fig EV2). The maintenance of CD57 and lack of elevation of CD90 / CD105 / CD73 and the same ability to differentiate into NCCs derivatives suggest that the homozygous $ALX1^{165F/165F}$ failed to progress through the process of NCCs differentiation despite multiple cell passages and are blocked into the progenitor state.

### Homozygous ALX1$^{165F/165F}$ NCC display a migration defect

During embryonic development, NCC migrate to specific locations in order to form the prominences that coalesce to shape the face. To investigate the migratory properties of the iPSC-derived NCC *in vitro,* a wound healing assay with a central clearing was used. A significant migration defect was observed in the homozygous $ALX1^{165F/165F}$ NCC when compared with control NCC (Fig 5A,

Movie EV1). Control NCC were able to migrate and fully cover the central clearing area of the wound healing assay after 24 h (recovery of 95.99 ± 3.22% of the surface area). In contrast, the homozygous $ALX1^{165F/165F}$ NCC covered less than half of the central clearing surface area (38.79 ± 3.22% for $ALX1^{165F/165F}$ NCC).

### ALX1$^{165F/165F}$ NCC show differences in BMP secretion

The family of BMP family of growth factors plays a critical role in NCC migration (Tribulo *et al,* 2003; Sato *et al,* 2005). This, in combination with the increased expression of *TWIST1* in $ALX1^{165F/165F}$ NCC, a known BMP inhibitor, led to us to hypothesize that $ALX1^{165F/165F}$ NCC might display abnormal levels of secreted BMP when compared to healthy control NCC (Hayashi *et al,* 2007). To test this hypothesis, the levels of secreted BMP in the culture medium of $ALX1^{165F/165F}$ and control NCC were measured via multiplex analysis. The concentration of BMP2 was found to be significantly reduced in control $ALX1^{165F/165F}$ NCC (11.9 ± 0.65 pg/ml) compared to control NCC (19.52 ± 0.9 pg/ml; $P < 0.05$; Fig 5B). In contrast, the BMP9 concentration was significantly increased in mutant $ALX1^{165F/165F}$ NCC (3.72 ± 0.85 pg/ml) compared to control NCC (0.25 ± 0.02 pg/ml). BMP4, BMP7, and BMP10 levels were undetectable.

To follow-up on the observed dysregulation of BMPs, we hypothesized that treatments to counteract BMP2 reduction or BMP9 elevation could result in an improved migration phenotype. The $ALX1^{165F/165F}$ NCC were treated with different concentrations of soluble BMP2, the BMP9 antagonist Crossveinless (CV2), or a combination of the two (Fig 5C, Fig EV3, Movies EV2 and EV4). Treatment with an increasing concentration of BMP2 from 10 to 50 ng/ml was able to restore the migration of homozygous $ALX1^{165F/165F}$ NCC in a dose-dependent manner. However, no difference was observed between 50 and 100 ng/ml of BMP2, suggesting a saturation effect (64.53 ± 3.17% for 50 ng/ml BMP2, and 67.31 ± 3.25% for 100 ng/ml BMP2).

Likewise, treatment with the BMP9 antagonist CV2 was able to partially rescue the migration defect of homozygous ALX1$^{165F/165F}$ NCC. The low dose of 10 ng/ml of CV2 did not show a significant effect on the migration defect of treated and untreated $ALX1^{165F/165F}$ NCC (37.5 ± 2.5% for 10 ng/ml CV2). As observed with BMP2, treatments with both 50 ng/ml and 100 ng/ml of CV2 were able to partially restore the ability of the homozygous $ALX1^{165F/165F}$ NCC to migrate, with no difference found between these two concentrations (57.6 ± 4.77% for 50 ng/ml of CV2, and 64.64 ± 3.36% for 100 ng/ml of CV2). Finally, we asked whether treatment with a combination of BMP2 and CV2 would exert an additive or synergistic effect to restore cell migration than single compound treatment.

**Table 1. Comparative FACS analysis of subject-derived *ALX1*$^{165F/165F}$ and control NCC at passages 1 and 4**

| | Passage 1 | | Passage 4 | |
|---|---|---|---|---|
| | Control (*ALX1*$^{165L/165L}$) | *ALX1*$^{165F/165F}$ | Control (*ALX1*$^{165L/165L}$) | *ALX1*$^{165F/165F}$ |
| CD57 | 45.6 ± 7.7% | 57.4 ± 7.3% | 6.78 ± 2.36% | 49.55 ± 17.53% |
| CD90 | 55.05 ± 5.24% | 63.6 ± 6.9% | 88.46 ± 2.05% | 67.8 ± 11.07% |
| CD105 | 51.7 ± 7.8% | 47.6 ± 11.4% | 96.31 ± 0.95% | 60.02 ± 12.66% |
| CD73 | 97.4 ± 1.08% | 79.2 ± 9.6% | 98.8 ± 0.44% | 83.95 ± 6.05% |

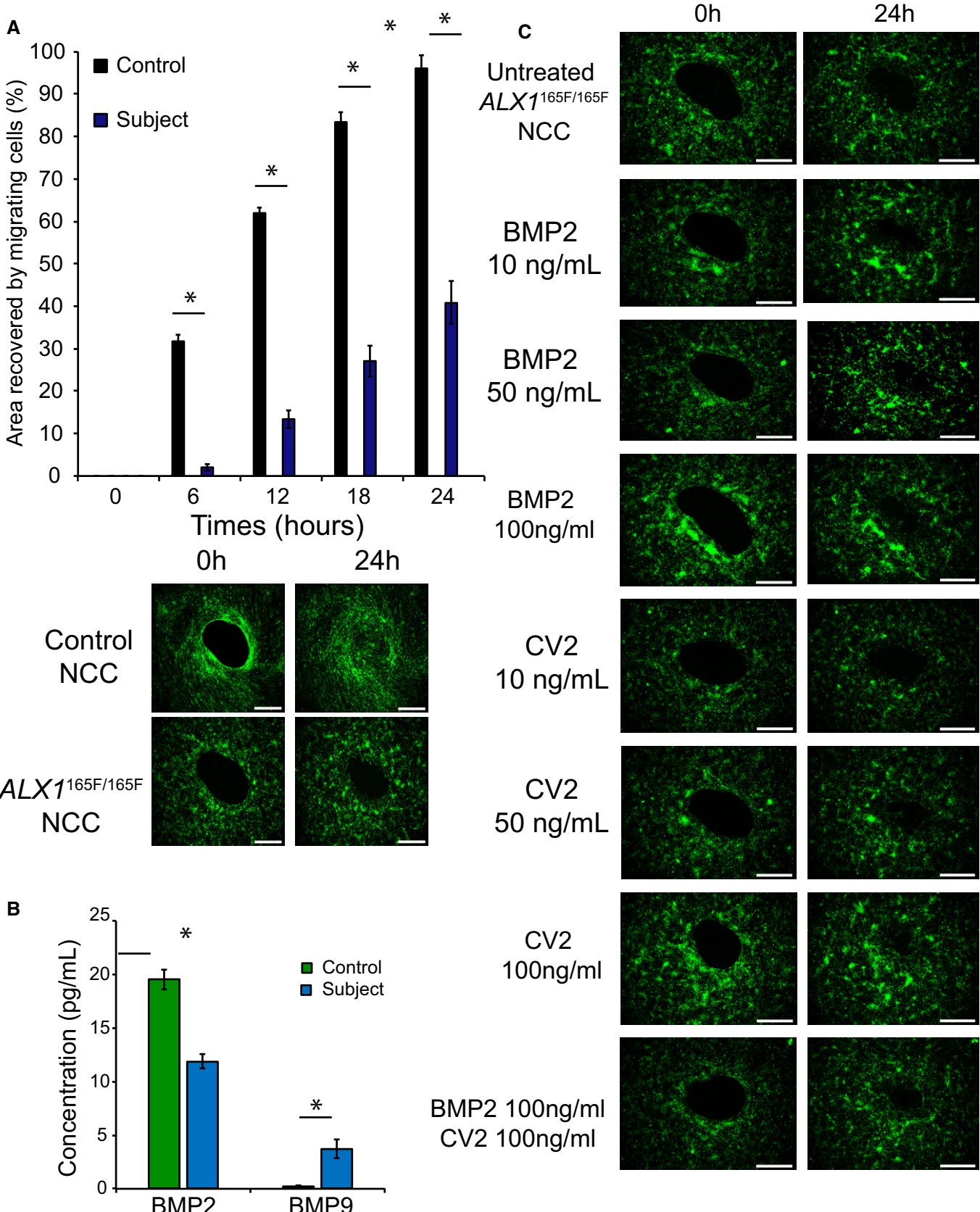

**Figure 5.**

**Figure 5.  $ALX1^{165F/165F}$ NCC show a migration defect and a difference in BMP secretion.**

A   Mutant $ALX1^{165F/165F}$ NCC (blue) exhibited a migration defect in timed coverage of the central clearing of the wound assay when compared with control $ALX1^{165L/165L}$ NCC (black). Data are presented as percent area recovery of the central circular clear area of the wound assay by migrating NCC at the end of a 24-h period. For fluorescent pictures, cells were stained in serum-free media containing 3.6 μM CellTracker Green CMFDA (Life Technologies) for 30 min at 37°C and allowed to recover for 30 min before starting the experiment. Images were acquired every 6 h using a Keyence BZ-X800 microscope. Surface area analyses and percentages of coverage were measured using ImageJ software (NIH). The NCC were monitored over 24 h. The data are represented as the average of the percentage of closure ± SEM. Scale bar = 200 μm. Data obtained of each clone from three independent experiments were pooled, and the mathematical mean was calculated. SEM was used to determine the standard error. To test statistical significance, an ANOVA test was performed. A $P$-value < 0.05 was considered to be statistically significant. *Significantly different from control NCC ($P$ < 0.0001).

B   Multiplex analysis of BMP2 and BMP9 in the supernatant of cultured NCC showed that $ALX1^{165F/165F}$ NCC (blue) secrete less BMP2 and more BMP9 compared to control $ALX1^{165L/165L}$ NCC (green). Data are represented as pooled mean ± SEM of three clones from each genotype. Statistical significance was determined with an ANOVA test. A $P$-value < 0.05 was considered significant. *Significantly different from control NCC ($P$ = 0.0424 for BMP2 and $P$ = 0.0192 for BMP9).

C   Addition of soluble BMP2 or CV2, a BMP9 antagonist, to the culture medium could partially rescue the migration defect of $ALX1^{165F/165F}$ NCC. At the beginning of the assay, 10, 50, or 100 ng/ml of soluble BMP2, CV2, or a combination of the two at 100 ng/ml each were added to the culture medium, and the cells were monitored over the next 24 h. The data are represented as the average of the percentage of closure ± SEM. Scale bar: 400 μm.

No additive effect was identified, as BMP2 and CV2 cotreatment at 100 ng/ml was able to rescue the migration defect phenotype of mutant $ALX1^{165F/165F}$ NCC at a similar level to what was observed with the individual treatments (73 ± 5.89% for BMP2 and CV2 cotreatment).

**Evaluation of *alx1* function in the zebrafish**

We and others previously showed that the zebrafish anterior neurocranium (ANC) forms from the convergence of the frontonasal prominence and the paired maxillary prominences (Wada, 2005; Eberhart, 2006; Dougherty *et al*, 2012). Since FND malformation is characterized by facial cleft affecting fusion of the frontonasal and maxillary structures, examination of the ANC morphology in zebrafish would be a sensitive assay for frontonasal development.

To generate an *in vivo* model of *alx1*, we employed CRISPR/Cas9-mediated targeted mutagenesis of the *alx1* locus in zebrafish. This approach produced a frameshift mutation harboring a 16-base pair (bp) deletion in exon 2 of *alx1* (NM_001045074), named $alx1^{uw2016}$ (Fig 6A, Fig EV4). The $alx1^{uw2016}$ mutation is likely to cause complete loss of function, since the encoded truncated protein lacks both the homeobox and transactivation domains. While the majority of $alx1^{uw2016}$ homozygous zebrafish developed normally and were viable as adults, approximately 5% displayed specific craniofacial defects (Fig EV5). Alcian blue staining of $alx1^{uw2016}$ homozygous larvae at 5 days post-fertilization (dpf) revealed that the posterior neurocranium and ventral cartilages and Meckel's cartilage were formed but smaller in size in a subset of zebrafish. In contrast, the ANC appeared dysmorphic, i.e., narrower in the transverse dimension and linear, rather than fan-shaped (Fig 6A). The chondrocytes of the ANC appeared cuboidal in mutant larvae, whereas wild-type ANC chondrocytes were lenticular and stacked in an intercalated pattern (Fig 6A).

The low penetrance of the ANC defect suggests the possibility of gene compensation by other *alx* family members (also see Discussion). To test this hypothesis, *alx* transcripts were quantified by qRT–PCR at several stages of embryogenesis in $alx1^{uw2016}$ homozygotes. We observed that *alx1* mRNA level was significantly decreased in $alx1^{uw2016}$ mutants across several time points, likely because the mutation triggers nonsense-mediated decay (Fig 6B; El-Brolosy *et al*, 2019). Consistent with transcriptional gene compensation, *alx3* and *alx4a* mRNA levels were significantly increased in the $alx1^{uw2016}$ mutant embryos compared to wild-type embryos ($P$ < 0.05 for 10 ss, 24 hpf and 36 hpf embryos for *alx3*; $P$ < 0.05 for 24 hpf and 36 hpf embryos for *alx4a*). These data suggest that *alx3* and *alx4a* functions may compensate for the loss of *alx1* during zebrafish development (Fig 6B).

Given the likely genetic compensation between different *alx* family members, we utilized a dominant-negative approach to interrogate the function of *alx* genes in embryonic development. It has

**Figure 6.  *Alx1* function in zebrafish.**

A   Dissected flatmount wild-type and $alx1^{-/-}$ zebrafish larvae craniofacial cartilages after Alcian blue staining, the anterior points to the left of the page in all images. The ventral cartilages appear normal, but the $alx1^{-/-}$ anterior neurocranium (ANC) appears narrow, with the midline element that is derived from the frontonasal NCC being absent. Meckel's cartilage (arrow, MC) is also diminutive. Scale bar: 200 μm.

B   Zebrafish *alx1* mutants (blue) show reduced detectable expression of *alx1* but increased expression of *alx3*, *alx4a* compared to wild-type controls (green). *alx4b* expression levels are similar between wild-type and $alx1^{-/-}$ lines. Data are represented as the mean of all pooled embryos from three different clutches. The RT–qPCR relative expression values were normalized to *elfa* and *18S* expression using the ΔΔCT method. Data from each clutch were pooled, and the mathematical mean was calculated. SEM was used to determine the standard error. To test statistical significance, an ANOVA test was performed. A $P$-value < 0.05 was considered to be statistically significant. Statistical significance denoted by *; $P$ < 0.0001 between WT zebrafish and $alx1^{-/-}$ at all measured time points; at 10 ss, 24 hpf and 36 hpf for *alx3*; and at 24 hpf and 36 hpf for *alx4a*. Refer to Table EV2 for $P$-values.

C   Dissected flatmount of zebrafish embryos injected with Alx1DN, after Alcian blue staining. The embryos developed an absence of the frontonasal-derived median portion of the anterior neurocranium (ANC) and a profound hypoplasia of Meckel's and ventral cartilages. In the most severely affected zebrafish, a nearly abrogated ANC was observed. Scale bar: 200 μm.

D   Lineage tracing experiments in control and Alx1DN mutant embryos revealed aberrant migration of anterior cranial NCC when *alx1* is disrupted. In the control animal, the anterior cranial NCC always migrate to contribute to the median portion of the ANC. In contrast, the anterior cranial NCC labeled in the Alx1DN animals fail to migrate to the median ANC, where the ANC structure is narrower and the labeled cranial NCC are found in an anterior and lateral ectopic location (white asterisks). Scale bar: 250 μm.

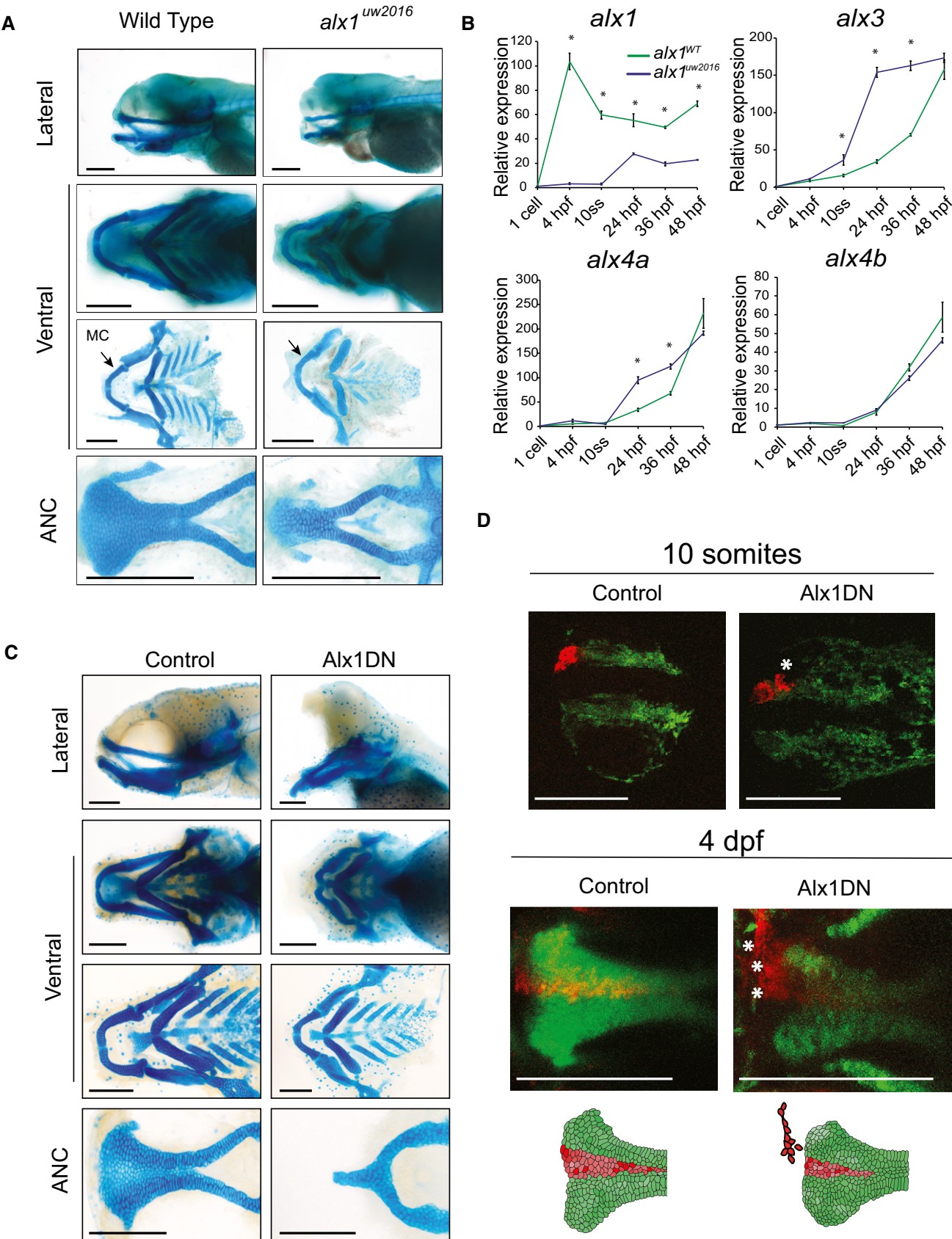

**Figure 6.**

been reported that ALX1 protein homodimerizes to be fully functional (Furukawa *et al*, 2002). In order to circumvent the genetic compensation by *alx3* and *alx4a*, a truncated form of *alx1* containing the N-terminal domain (Alx1DN) was generated. The Alx1DN truncation contains the DNA-binding homeodomain but is missing the transactivation domain (Herskowitz, 1987). Additionally, an alternative truncated protein that lacks the DNA-binding homeodomain and contains the transactivation domain, termed Alx1CT, was generated. We reasoned that, if a truncated Alx1 protein can occupy its binding sites but fail to dimerize or associate with its coactivators, it may function in a dominant-negative manner.

When Alx1DN was overexpressed by mRNA injection in wild-type zebrafish embryos, embryos displayed significant craniofacial defects (Fig 6C), while Alx1CT-expressing embryos developed normally. On closer inspection with Alcian blue staining, Alx1DN-expressing embryos showed complete abrogation of the frontonasal-derived median section of the ANC and Meckel's cartilage, a similar but more severe phenotype as expected due to the interference with all alx proteins than that observed in *alx1^{uw2016}* mutant embryos. Both *alx1^{uw2016}* and the Alx1DN mutant phenotypes suggest that alx1 regulates the migration of the anterior frontonasal NCC stream that contributes to the median portion of the ANC.

### Lineage tracing reveals migration defect of NCC

To elucidate whether anterior cranial NCC migration is specifically affected in Alx1DN embryos, lineage tracing of the migrating NCC was carried out using the *Tg(sox10:kaede)* reporter line (Wada, 2005; Eberhart, 2006; Dougherty *et al*, 2012; McGurk *et al*, 2014). The anteromost stream of NCC in wild-type and Alx1DN injected embryos was labeled at the 10-somites stage and followed to 96 h post-fertilization (Fig 6D). In the wild-type embryos, at 4 days post-fertilization (dpf), the NCC were able to migrate into the frontonasal region of the palate (Fig 6D). In contrast, the anterior cranial NCC of Alx1DN injected embryos were unable to reach their final location of the median ANC. Instead, the NCC in the AlxDN injected zebrafish were found in an ectopic anterior location outside of the frontonasal domain (Fig 6D, Movie EV3). While it is possible that increased cell death and altered cell division that were observed in the iPSC model are also operating here in the embryo, these cellular derangements are likely minor contributors to explain the ectopic localization of the cranial NCC, whereas altered cell migration is the dominant mechanism.

## Discussion

We report the identification of a novel missense variant of human *ALX1* associated with FND. Analysis of this putative loss-of-function variant in patient-derived iPSC and NCC showed a lack of cellular maturation, an increase in apoptosis, and a migration defect. We identified an overexpression of neural plate border specifiers in subject-derived cells, and an imbalance of BMP levels which, when addressed, was capable of mitigating the migration defect discovered in the subject's NCC. A delay of NCC migration was also recognized as key to the morphologic consequences of a loss of alx1 in zebrafish models. We could identify genetic compensation between different members of the alx gene family.

### Human genetics of *ALX1*

Genes of the *ALX* family encode homeodomain transcription factors and are associated with craniofacial malformations. Like other members of this family, the ALX1 protein is composed of two main functional domains: the N-terminal portion containing the DNA-binding homeodomain with two nuclear localization signals, and the C-terminal portion containing an OAR/aristaless domain required for transactivation and protein interaction (Furukawa *et al*, 2002). In this study's pedigree, a novel missense variant p.L165F within the conserved homeodomain was identified. Leucine is an aliphatic, branched amino acid, while phenylalanine is an aromatic, neutral, and nonpolar amino acid. Due to properties of leucine, the substitution itself is likely disruptive to helix II in the DNA-binding element of the homeodomain within which it resides. Disruptive leucine to phenylalanine substitutions have been described in a number of published, genotyped disorders (Miller *et al*, 1992; Gomez & Gammack, 1995). Pathogenic missense variants within the homeodomains of both *ALX3* and *ALX4* have previously been identified as the causes of FND types 1 and 2, respectively (Wuyts *et al*, 2000; Kayserili *et al*, 2009; Twigg *et al*, 2009).

Pathogenic *ALX1* gene variants in FND have been reported in two case studies in the literature. The first study described two families (Uz *et al*, 2010). In the first, three siblings of consanguineous parents were described to suffer from a midline defect of the cranium, bilateral extreme microphthalmia, bilateral oblique cleft lip and palate, a caudal appendage in the sacral region, and agenesis of the corpus callosum. A homozygous 3.7 Mb deletion spanning *ALX1* was identified in all affected subjects. In a second family, one affected subject was born with a midline defect of the cranium, bilateral microphthalmiamicrophthalmia, bilateral oblique cleft lip and palate, and a thin corpus callosum. A donor splice-site mutation c.531+1G>A of *ALX1,* homozygous in the child and heterozygous in the parents, was identified to be the likely cause of the child's phenotype.

None of the affected subjects from the pedigree reported in this study demonstrated midline defects of the cranium or a cerebral phenotype. This is in spite of the fact that the missense mutation of the affected subjects in our study lies just proximal to that of family 2, within helix II, within the homeodomain.

A second case report described one family with FND (Ullah *et al*, 2016). It reported on four children born of consanguineous parents that presented with a broad nasal root, smooth philtrum, and mouth protrusion. An *ALX1* gene variant c.661-1G>C was found to be heterozygous in the parents and homozygous in the affected children. The skipping of the exon via alternative splicing likely resulted in a mutant protein with some residual function, explaining the relatively mild phenotype.

### The *ALX* gene family: *ALX1, ALX3,* and *ALX4*

The *ALX* gene family consists of three members: *ALX1, ALX3,* and *ALX4* (McGonnell *et al*, 2011). In humans, mutations of each *ALX* gene have been associated with craniofacial malformations of the frontonasal-derived structures with variable phenotypic severity (Wu *et al*, 2000; Wuyts *et al*, 2000; Mavrogiannis *et al*, 2001; Kayserili *et al*, 2009; Twigg *et al*, 2009; Uz *et al*, 2010). FND is a descriptive term that broadly describes a number of malformations

of the midface. Previously classified based on appearance (see Tessier, Sedano, De Myer classifications), FND related to variants within the *ALX* gene family has recently been reordered on the basis of genetics: Type I is caused by mutations of *ALX3*; type 2 is caused by mutations of *ALX4*; and type 3 is caused by mutations of *ALX1*. FND types 1 and 2 appear milder than type 3, frequently presenting with altered appearance of the nasal soft tissue (Twigg *et al*, 2009).

In mouse and zebrafish, *Alx1, Alx3,* and *Alx4* have been shown to be expressed in spatiotemporally restricted regions of the craniofacial mesenchyme (Zhao *et al*, 1994; Qu *et al*, 1997; ten Berge *et al*, 1998; Beverdam & Meijlink, 2001; Dee *et al*, 2013; Lours-Calet *et al*, 2014). Evidence of gene compensation has previously been reported in animal studies (Beverdam *et al*, 2001; Dee *et al*, 2013). In studies of sea urchins, *Alx4* was shown to be directly regulated by Alx1 (Rafiq *et al*, 2012; Khor *et al*, 2019). The question remains how the different *ALX* family members regulate craniofacial development, through transcriptional activation or repression of shared and unique target genes.

### iPSC-derived NCC for craniofacial disease modeling

Most craniofacial structures are derived from a transient multipotent embryonic population called NCC. The NCC progenitors give rise to a wide variety of cell lineages including peripheral neurons, melanocytes, and craniofacial mesenchyme (Betancur *et al*, 2010; Stuhlmiller & Garcia-Castro, 2012). NCC exhibit a restricted expression of *ALX1* in the rostral domain during early developmental stages (Zhao *et al*, 1996; Dee *et al*, 2013). Cellular and genetic mechanisms that drive frontonasal NCC formation are poorly understood. In order to gain insight into the functional consequences of the clinically pathogenic *ALX1* gene variant identified in this study's pedigree, iPSC were differentiated into NCC.

While a number of sophisticated protocols using chemically defined mediums and a combination of adherent and/or suspension culturing approaches have been published in recent years, no consensus has been established on a number of controversial issues (Bajpai *et al*, 2010; Leung *et al*, 2016; Tchieu *et al*, 2017).

First, the definition of what a NCC in fact is remains based entirely on the transcriptomic profiling performed in animal studies. While we succeeded in identifying distinctive differences between the *ALX1*[165F/165F] NCC and healthy controls, the results suffer from an obvious limitation: a lack of understanding which stage of development the NCC represent. The central challenge of the work with iPSC models of human disease remains the lack of available human transcriptomic cell data to allow for an understanding which stage of development is modeled by the cellular lineages derived. NCC are characterized *in vitro* by the expression of markers identified to be specific to this cell population, namely P75, CD57, CD90, CD73, and CD105 (Minarcik & Golden, 2003; Billon *et al*, 2007; Kawano *et al*, 2017) as well as their multilineage differentiation ability. NCC formation is a stepwise process coordinated by a spatiotemporally specific gene expression pattern. In this study, a putative loss of *ALX1* function did not impair NCC differentiation itself or the ability of NCC to differentiate into multiple cell lineages. Rather, it appeared that the clinically pathogenic *ALX1*[165F/165F] variant maintained the NCC in a precursor state. While control cells progressively became craniofacial mesenchymal cells by CD57 downregulation and increases in MSC associated marker expression,

*ALX1*[165F/165F] NCC did not undergo changes of their morphology or show a transition of progenitor to mesenchymal markers. Second, a lack of a 3D representation of NCC migration *in vitro* based studies force scientists to either transplant human iPSC-derived NCC into model organisms or retain a 2D system of representation (Bajpai *et al*, 2010; Okuno *et al*, 2017). We focused on the validation of the findings in human iPSC in zebrafish. Third, the development of craniofacial mesenchyme is a product of the interactions of derivatives of all three germ layers. This left the role of *ALX1* in other developmental derivatives unexplored in this study.

To allow for some insight into the expression profile of key NCC markers during the *in vitro* differentiation process, we surveyed relative expression via qPCR every 2 days throughout our differentiation protocol. We found the greatest differences between the unaffected father and the affected children in the expression of *PAX7, PAX3, DLX5, SNAI2*, and *TWIST1*. *ALX1* has been described as a transcription modulator capable of both activating and repressing target gene expression, adapting its activity to different cell types and environments (Gordon *et al*, 1996; Cai, 1998; Damle & Davidson, 2011). Its activity as a repressor, for example, has been documented with prolactin, as *ALX1* binds directly to the prolactin promoter (Gordon *et al*, 1996). In this study, all of the genes were substantially upregulated in the affected children, with the greatest changes found in the earlier time points of differentiation. ALX1 appears to play the role of a transcriptional repressor in NCC-based craniofacial development.

Neural crest cells delamination and migration depend on signals from the surrounding epidermis, including BMPs, which induce expression of neural plate border specifier genes such as *PAX3, TFAP2a, MSX1/2* or *ZIC1* (Tribulo *et al*, 2003; Sato *et al*, 2005; Garnett *et al*, 2012; Dougherty *et al*, 2013; Simoes-Costa & Bronner, 2015). Fine temporospatial regulation of the level of these signaling molecules is critical to allow for delamination and migration of NCC craniofacial development. BMPs belong to the transforming growth factor beta (TGFβ) superfamily and can be divided into several subcategories based on molecular similarities. The two BMPs showing dysregulation in this study, BMP2 and BMP9, belong to different subcategories which exhibit different expression patterns and receptors. BMP2 was identified as an important factor in migratory NCC development, with a depletion of mobile NCC in knockout models in transgenic mice resulting in hypomorphic branchial arches. (Kanzler *et al*, 2000). In a complementary mouse model, BMP2 increased migration of NCC when added as a supplement to culture medium (Anderson *et al*, 2006). BMP2 was also found to be required for the migration of NCC in the enteric nervous system in the zebrafish and to be significantly decreased in the gut of patients affected by Hirschsprung's disease, a disease characterized by deficient enteric NCC migration (Huang *et al*, 2019a). BMP9 on the other hand was shown to be required for tooth development in mice (Huang *et al*, 2019b). It was previously identified as a potent inducing factor of osteogenesis, chondrogenesis, and adipogenesis during development (Luther *et al*, 2011; Lamplot *et al*, 2013). Opposed to other BMPs, including BMP2, BMP9 was found to be resistant to feedback inhibition by BMP3 and noggin (Wang *et al*, 2013).

The relationship of BMP2 and BMP9 in NCC development, migration, and differentiation has yet to be examined. Why BMP2 and BMP9 appeared to play antagonistic roles in the NCC modeling of *ALX1*-related FND presented in this study remains unclear. On the

basis of the qPCR data and the multiplex assay revealing a decrease in BMP2 and an increase in BMP9 in NCC supernatant, we hypothesized that a lack of fully functional ALX1 may account for the overexpression of neural plate border specifiers, and the change of BMP signaling. In substituting or repressing BMP2 and BMP9, respectively, an almost complete rescue of the migration defect of the mutant *ALX1*[165F/165F] NCC was achieved. Pretreatment of subject-derived NCC could perhaps result in a complete rescue of migration.

### Animal models of *ALX1*-related FND

Studies in sea urchins have contributed meaningful knowledge to the regulatory functions of Alx1 as a transcription factor. In the sea urchin *Strongylocentrotus pupuratus*, the *alx1* gene was found to activate itself in a self-regulatory loop at lower levels. Once its level exceeds a certain threshold, *alx1* reverses its activity and becomes a repressor of its own transcription (Damle & Davidson, 2011). As a transcription factor, alx1 was found to be essential for the regulation of epithelial–mesenchymal transition, a process of great importance for the ability of NCC to delaminate and initiate migration (Ettensohn *et al*, 2003).

The specific role of *Alx1* in craniofacial development was investigated in different animal models. Targeted gene ablation of *Alx1* in mice resulted in neural tube closure defects in the majority of the pups, a phenotype not observed in any reported case report of *ALX1*-related FND type 3 (Zhao *et al*, 1996). A previously published morpholino knockdown of *alx1* in zebrafish suggested that the gene is essential for the migration of NCC into the frontonasal prominence, with a disorganization of NCC in the frontonasal stream, and reduction both in the number of NCC and its cellular projections (Dee *et al*, 2013). A major weakness of morpholino gene disruption is nonspecific or off target effects. This study utilized germline *alx1* mutant allele to investigate the effect of *alx1* loss-of-function, complemented by a dominant-negative disruption approach to address gene compensation of other *alx* family members. These approaches corroborate a requirement for alx1 in median ANC morphogenesis, corresponding to formation of the midface in humans.

In summary, this work describes a novel *ALX1* gene variant associated with FND. Using complementary human iPSC and zebrafish models, this study showed that *ALX1* is required for coordinated NCC differentiation and migration. Discordance of NCC differentiation from cell migration during midface morphogenesis results in FND. Future work will be directed at identifying *ALX1* downstream targets and characterize the ALX regulated pathways in craniofacial development.

# Materials and Methods

### Approvals to perform research with human samples and zebrafish

The collection of human blood and discard specimens, genome sequencing, and generation of iPSC was approved by the Institutional Review Board of Partners Healthcare (IRB No. 2015P000904). Informed consent was obtained from the parents of the patients prior to all sample collections. All experimental protocols using

zebrafish were approved by the Animal Care and Use Committees of Massachusetts General Hospital (IACUC No. 2010N000106) and the University of Wisconsin and carried out in accordance with institutional animal care protocols.

### iPSC and EB generation

Peripheral blood mononuclear cells were isolated using whole blood from two individuals (subjects 5 and 6: *ALX1*[165F/165F]), the unaffected father (subject 1: *ALX1*[165L/165F]), and three unrelated healthy individuals (controls: *ALX1*[165L/165L]). Samples were diluted in an equal volume of PBS and gently transferred to a tube containing 4 ml of FICOLL. After centrifuging the sample for 10 min at 350 *g*, the FICOLL-plasma interface containing the PBMCs was recovered and washed several times in PBS. After 24 h of recovery in StemPro-34 SFM medium (Invitrogen) supplemented with 100 ng/ml Stem Cell Factor (SCF, PeproTech, Rocky Hill, NJ), 100 ng/ml Fms-related tyrosine kinase 3 ligand (Flt3L, PeproTech), 20 ng/ml Interleukin-3 (IL-3, PeproTech) and 20 ng/ml IL-6 (PeproTech), 1 million PBMC were processed using the CytoTune-iPS 2.0 Sendai Reprogramming Kit (Invitrogen, Carlsbad, CA), following manufacturers instruction, for iPSC generation. One million PBMCs were infected with three different Sendai Viruses containing the Yamanaka reprogramming factors, *OCT4, SOX2, KLF4,* and *c-MYC*, in StemPro-34 SFM medium supplemented with cytokines. Starting on day 21, individual iPSC clones were picked based on morphologic criteria. Subsequently, the iPSC were maintained in StemFlex medium and passaged 1–2 a week using ReLSR (STEMCELL Technologies, Vancouver, BC, Canada) dissociation buffer. Since iPSC can exhibit genetic instability after reprogramming, the clones were expanded up to passage 10 before characterizing each cell line. The genetic stability of the cells was assessed analyzing copy number variants through genome-wide microarray analysis (Thermo Fisher). Epigenetic differences were controlled for in a limited manner by ensuring that all major experiments were performed in both biologic and technical triplicate at the identical passage number.

To form embryoid bodies (EB), iPSC were harvested using ReLSR dissociation buffer and clumps were transferred to a low adherent 6-well plate in differentiation medium containing 80% DMEM-F12, 20% Knock out Serum Replacer (Invitrogen), 1 mM nonessential amino acids, 1 mM Penicillin/Streptomycin, and 100 μM 2-mercaptoethanol. The medium was changed daily. After 14 days of differentiation, cells were recovered for RNA extraction and subsequent qPCR analysis of markers of the ectoderm, endoderm, and mesoderm.

### Derivation of NCC and multilineage differentiation

In order to derive NCC, a previously published protocol for mesenchymal differentiation was adapted (Pini *et al*, 2018). iPSC medium was replaced by NCC-inducing medium containing DMEM-F12, 10% fetal bovine serum (FBS), 1 mM sodium pyruvate, 1 mM Penicillin/Streptomycin, 1 mM nonessential amino acid, 110 μM 2-mercaptoethanol, and 10 ng/ml epidermal growth factor (EGF). The medium was changed every 2 days. After 1 week, cells were recovered using 0.25% trypsin-EDTA and transferred to new cultureware for an additional week. Following this, cells were harvested,

phenotypically characterized by flow cytometry for their expression of NCC markers and assayed for their mesenchymal differentiation ability.

Schwann cell differentiation was performed as previously described (Kawano *et al*, 2017). NCC were plated on glass coverslips in 24-well tissue culture plates ($0.2 \times 10^5$ cells per well) in neuronal differentiation medium consisting of a 3:1 ratio of DMEM-F12 and neurobasal medium supplemented with $0.25\times$ B-27, 1 mM glutamine, and 1 mM Penicillin/Streptomycin for 5 weeks. The medium was changed weekly. At the end of the differentiation, cells were fixed in 4% formaldehyde and analyzed by immunohisto-chemistry for S100B (Thermo Fisher, Waltham, MA) and GFAP expression (Abcam, Cambridge, United Kingdom).

Adipocyte and chondrocyte differentiation was performed as previously described (Pini *et al*, 2018). Adipogenesis was investigated using the StemPro adipogenesis differentiation kit (Life Technologies, Carlsbad, CA). NCC were seeded at $5 \times 10^4$ per well, in a 24-well plate, and cultured for 2 weeks, in complete adipogenesis differentiation medium. Lipid deposits were observed following staining with Oil Red O (MilliporeSigma, St. Louis, MO), according to manufacturers' instructions. After washing, cells were counterstained with Mayer's hematoxylin.

Chondrogenic differentiation was performed using the StemPro chondrogenesis differentiation kit (Life Technologies). NCC were seeded in a 12-well plate, in aggregates containing $8 \times 10^4$ cells, in 5 µl of NCC medium, and placed in a 37°C, 5% $CO_2$ incubator for 1 h. Following this, the NCC medium was replaced by chondrogenesis differentiation medium, and cultured for 20 days. The medium was changed once a week. Chondrogenic matrix formation was observed following Alcian blue, Safranin O and Toluidine Blue staining.

Osteoblast differentiation was performed using the StemPro osteogenesis differentiation kit (Life Technologies). NCC were plated in 12-well tissue culture plates ($5 \times 10^5$ cells per well) in osteogenesis differentiation medium for 14 days. At the end of the differentiation, the presence of mineralized nodules was assessed using Alizarin Red S, Von Kossa (silver nitrate) and Alkaline Phosphatase staining.

Images were acquired using the RETIGA OEM fast camera and Qcapture software (Teledyne QImaging, Surrey, BC, Canada).

## Genomic DNA extraction and sequencing

Genomic DNA was extracted with the REDExtract-N-Amp tissue PCR kit (MilliporeSigma), following the manufacturer's instructions. Sanger sequencing of *ALX1* to ensure *ALX1* sequence integrity in all iPSCs clones was carried out as previously described (Umm-e-Kalsoom *et al*, 2012). The four *ALX1* exons encoding the open reading frame were amplified using the CloneAmp HiFi PCR premix (Takara Bio Inc., Kusatsu, Shiga, Japan) and exon-specific *ALX1* oligonucleotides. All exon-specific PCR products were purified using the Qiaquick PCR purification kit (Qiagen, Hilden, Germany) prior to sequencing.

## Whole-exome sequencing and analysis

Whole-exome sequencing (WES) of the affected subjects 3 and 4, an unaffected sibling, and the parents was performed and analyzed assuming a recessive mode of inheritance given the presence of multiple affected siblings. Three compound heterozygous variants and one homozygous recessive variant were identified in the affected siblings (*ALX1* c.493C>T). This variant was predicted to be causative of the phenotype based on known gene function, the previously identified role of *ALX1* in frontonasal development, and the effect of the variant (substitution of phenylalanine for the highly conserved leucine in a DNA-binding domain). Polymorphism Phenotyping v2, Sorting Intolerant from Tolerant, MutationTaster, and Functional Analysis through Hidden Markov Models (v2.3) were used for functional variant consequence prediction (Lowe, 1999; Adzhubei *et al*, 2010; Schwarz *et al*, 2014; Shihab *et al*, 2014). The gnomAD platform was used to identify any other missense variants at the location identified in the subjects (preprint: Karczewski *et al*, 2019). Clustal Omega was used for multiple sequence alignment (Sievers *et al*, 2011). Domain Graph was used to create the annotated schematic diagrams of *ALX1* and ALX1 (Ren *et al*, 2009).

## RNA extraction and processing

RNA was isolated using the RNAeasy Plus mini kit (Qiagen), following the manufacturer's recommendations. One µg of RNA was reverse-transcribed using the SuperScript III first-strand synthesis system (Thermo Fisher). All PCR reactions on cDNA were performed using the GoTaq DNA polymerase (Promega, Madison, WI) unless otherwise noted. For zebrafish RNA extraction, 24 hpf Tübingen zebrafish embryos were harvested and homogenized using a micropestle in TRIzol reagent (Thermo Fisher), following manufacturer's instructions. Total RNA was then purified using phenol-chloroform. One µg of total RNA was reverse-transcribed using the SuperScript III First-Strand Synthesis Kit (Thermo Fisher), following manufacturer's recommendations.

## Flow cytometry analysis

Neural crest cells were harvested and suspended in FACS buffer solution consisting of PBS with $Ca^{2+}$ and $Mg^{2+}$, 0.1% bovine serum albumin (BSA), and 0.1% sodium azide. Approximately $2 \times 10^5$ cells were incubated with the desired cell surface marker antibodies or isotype controls at 4°C for 15 min. Specific antibodies for CD90, CD73, CD105, and CD57 (BD Biosciences, San Jose, CA), and isotype control immunoglobulin IgG1 (BD Biosciences) were used for labeling. Antibodies were diluted in FACS buffer. After three washes in FACS buffer, samples were fixed in 0.4% formaldehyde and processed using an LSR II flow cytometer (BD Biosciences). The data acquired were analyzed using FlowJo software (FlowJo, LLC).

## Immunohistochemical analysis of iPSC

Cells were fixed with 4% formaldehyde in PBS for 15 min at room temperature, permeabilized with 1% saponin in PBS, and blocked using 3% BSA in PBS for 30 min at room temperature. The cells were then incubated with the primary antibodies for 3 h at room temperature. The following primary antibodies and dilutions were used: rabbit anti-OCT4 (1:100, Life Technologies), mouse anti-SSEA4 (1:100, Life Technologies); rat anti-SOX2 (1:100, Life Technologies), mouse anti-TRA-1-60 (1:100, Life Technologies), rabbit

anti-GFAP (1:500, Abcam), and rabbit anti-S100B (1:500, Thermo Fisher). The cells were then incubated with the secondary antibodies (1:1,000, MilliporeSigma) for 1 h at room temperature, washed with PBS, and counterstained with DAPI (MilliporeSigma). Secondary antibodies were Alexa 594 donkey anti-rabbit, Alexa 488 goat anti-mouse, and Alexa 488 donkey anti-rat and Alexa 594 goat anti-mouse (Thermo Fisher). Images were acquired using the RETIGA OEM fast camera and Qcapture software (Teledyne QImaging).

### Staining of iPSC and mesenchymal NCC derivatives

Alkaline phosphatase activity was measured using the leukocyte alkaline phosphatase staining kit (MilliporeSigma), following the manufacturer instructions. Cells were first fixed using a citrate/acetone/formaldehyde solution for 30 s, washed several times, and stained with Fast Blue for 30 min. After further washing, these cells were counter stained with Mayer's hematoxylin. Alizarin Red S., Von Kossa, Alcian Blue, and Toluidine Blue staining were performed as previously described (Pini *et al*, 2018). Cells were first fixed in 4% formaldehyde at room temperature for 15 min. Following a wash, cells were incubated in either 1% Alizarin Red, 1% Silver nitrate, 0.1% Toluidine Blue, 0.02% Alcian Blue, or 0.1% Safranine O solution. For Von Kossa staining, cells were exposed to UV light until dark staining appeared. Images were acquired using the RETIGA OEM fast camera and Qcapture software (Teledyne QImaging).

### Quantitative and nonquantitative polymerase chain reaction

Real-time PCR assays were conducted on a StepOnePlus real-time PCR system, using PowerUp SYBR Green Master Mix (Applied Biosystems, Waltham, MA). Transcript expression levels were evaluated using a comparative CT process ($\Delta\Delta$CT) with human *RPLP0* and *GAPDH* used as reference genes. For zebrafish, *elfa* and *18S* were used as reference genes. Specific primers were used for amplification as noted (Table EV1).

### Apoptosis assay

$2 \times 10^5$ cells were incubated for 30 min in the dark in 1× Fixable Viability Dye (FVD, Invitrogen) solution. After two washes in FACS buffer and one wash in binding buffer, cells were incubated 10–15 min in 1× Annexin V (BioLegend, San Diego, CA) solution in binding buffer composed of 0.1 M HEPES (pH 7.4), 1.4 M NaCl, and 25 mM $CaCl_2$. After one wash in binding buffer, cells were suspended in 200 µl of binding buffer and immediately processed using an LSR II flow cytometer (BD Biosciences) and analyzed using FlowJo software (FlowJo LLC, Ashland, OR). Apoptosis was induced by placing the cell suspension in a water bath at 55°C for 10 min.

### Wound healing assay and analysis

Migration was investigated using the Radius™ 48-Well Cell Migration Assay (Cell Biolabs, Inc., San Diego, CA), following manufacturer's instructions. Control or *ALX1*[165F/165F] NCC ($1 \times 10^5$ cells/well) were plated in a 48-well plate containing a Radium™ Gel spot. Before beginning the migration assay, cells were washed three times

with medium and incubated with gel removal solution for 30 min at 37°C. Following three subsequent washes in medium, the NCC were placed in a culture chamber for live cell imaging at 37°C and 5% $CO_2$. Rescue experiments were performed through the addition of soluble BMP2, CV2, or a combination of the two at a concentration of 10, 50, or 100 ng/ml to the medium at the beginning of the assay. For fluorescent pictures, cells were stained in serum-free media containing 3.6 µM CellTracker Green CMFDA (Life Technologies) for 30 min at 37°C and allowed to recover for 30 min before starting the experiment. All images were acquired using a Keyence BZ-X800 microscope. The time-lapse film was made by acquiring images every 15 min for 24 h. The fluorescent images were acquired every 6 h. Surface area analyses and percentages of recovery were measured using ImageJ software (NIH, Bethesda, MD).

### Multiplex analysis of BMP concentration

The concentration of the BMP family in the supernatant of *ALX1*[165F/165F] NCC was measured using a bead-based multiplex array (Forsyth Institute, Cambridge, MA). Manufacturers' protocols were followed for all panels. Reagents were prepared as per kit instructions. Assay plates (96-well) were loaded with assay buffer, standards, supernatant from the *ALX1*[165F/165F] NCC, and beads and then covered and incubated on a plate shaker (500 rpm) overnight at 4°C. After primary incubation, plates were washed twice. Following this, the detection antibody cocktail, consisting of BMP-specific antibody with biotin (1:10) and the detection antibody streptavidin conjugated with PE (1:25), was added to all wells; the plates were covered and left to incubate at room temperature for 1 h on a plate shaker. After the incubation, streptavidin-phycoerythrin fluorescent reporter was added to all wells, and the plate was covered and incubated for 30 min at room temperature on a plate shaker. Plates were then washed twice, and beads were resuspended in sheath fluid, placed on shaker for 5 min, and then read on a Bio-Plex® 200 following manufacturers' specifications and analyzed using Bio-Plex Manager software v6.0 (Bio-Rad, Hercules, CA).

### Plasmid construct generation

The In-Fusion Cloning Kit (Takara) and the In-Fusion Cloning primer design tool were used for primer design. Tübingen zebrafish *alx1* was amplified via PCR. Zebrafish *alx1* as cloned into the SpeI and PacI (NEB, Ipswich, MA) restriction sites of pCS2 + 8 (Promega). The subsequent reaction product was used to transform One Shot TOP10 competent cells (Thermo Fisher) or Stellar competent cells (Takara).

For the generation of truncated *alx1* constructs, the genes were divided into N-terminal and C-terminal sections, with aa181 being designated as the beginning of the C-terminal portion in zebrafish *alx1*.

For all plasmid constructs, individual clones were picked, DNA purified (Qiagen), and validated using Sanger and Next Generation whole plasmid sequencing.

### CRISPR-Cas9 directed mutagenesis of zebrafish

A CRISPR site on exon 2 of the *alx1* gene was selected using the Burgess lab protocol (Varshney *et al*, 2015, 2016). The single guide

RNA (sgRNA) targeting this site (sequence: GGAGAGCAGCCTG-CACGCGA), and Cas9 or nCas9n mRNA, were prepared as previously described (Gagnon *et al*, 2014; Shah *et al*, 2016a,b). Genetically defined wild-type (NIHGRI-1; LaFave *et al*, 2014) embryos were injected at the one-cell stage with 50–100 pg of sgRNA and 360–400 pg of Cas9 or nCas9n mRNA. Adult F0 animals were intercrossed to produce the F1 generation. F1 mutant carriers were identified by PCR using forward primer CGTGACT-TACTGCGCTCCTA and reverse primer CGAGTTCGTC-GAGGTCTGTT. The PCR products were resolved on a MetaPhor gel (Lonza, Basel, Switzerland) and sequenced. A frameshift allele was identified: a deletion of 16 nucleotides, termed $alx1^{uw2016}$ (Fig EV4).

### Alx1DN expression in zebrafish embryos

The validated Alx1DN (N-terminal portion of protein product containing homeodomain and nuclear localization domains) clones in pCS2+8 were purified via miniprep (Qiagen) alongside a control (C-terminal portion containing transactivation domain) and digested using NotI (NEB), before being gel purified using the Zymoclean Gel DNA Recovery Kit (Zymo Research, Irvine, CA). Five hundred microgram of purified, digested plasmid DNA served as the input for the mMessage mMachine SP6 Transcription Kit (Thermo Fisher). The resulting mRNA was then further purified using the RNeasy Mini Kit (Qiagen) and frozen in 100 ng/μl aliquots at −80°C. mRNA overexpression was accomplished using microinjections. mRNA stock aliquots were first diluted to the desired concentration with 0.125% Phenol Red in ultrapure water (Invitrogen). A 2 nl drop was then injected into fertilized Tübingen embryos at the single cell stage. At 4 h post-fertilization (hpf), all unfertilized and visibly damaged embryos were removed.

### Alcian blue staining

All injected and uninjected zebrafish were incubated at 28.5°C for 5 dpf in E3 buffer with 0.0001% methylene blue. At 4 days post-fertilization (dpf), injected and uninjected embryos were fixed overnight at 4°C in 4% formaldehyde, washed stepwise with 1 × PBS and 50% EtOH in PBS before being stained in a solution of 0.02% Alcian blue, 70% EtOH, and 190 mM $MgCl_2$ overnight at room temperature on a rotating platform. Following this, embryos were washed with $ddH_2O$ before being bleached in a solution of 0.9% $H_2O_2$, 0.8% KOH, and 0.1% Tween20 for 20 min. Stained embryos were then imaged in 4% methylcellulose in E3 solution and stored in 4% PFA at 4°C. Images were captured using a Nikon DS-Fi3 digital camera.

### Lineage tracing

The application of *Tg:sox10:kaede* in cell labeling has previously been reported, where photoconversion of the kaede protein from green to red in selected cells under confocal microscopy can be used to follow distinct NCC migration patterns across time.

Alx1DN injected or control *sox10:*KAEDE transgenic embryos were imaged using a Leica SP8 confocal microscope at 19 to 20-somite stage to identify neural crest structures in a manner previously described (Dougherty *et al*, 2012, 2013). The most distal population of the migrating stream of cranial neural crest cells was excited for 15 s for photoconversion with the FRAP module

### The paper explained

#### Problem
The causes of malformations of the human face remain poorly understood. This lack of understanding results in limited treatment and counseling options, specifically in families affected by malformations linked to a genetic cause. One such gene is called *ALX1*. This study aimed to understand the role of this gene in the development of the face and the effect of mutations of the gene in the genesis of malformation. To do so, we reprogrammed blood cells from children affected by *ALX1*-related malformations of the face into stem cells which allow us to retrace development. Additionally, we created a disruption of the gene in zebrafish in order to model the malformation in an animal and understand the role of the gene in development more broadly.

#### Results
*ALX1* was found to be crucial to the development of a cell population which exists only during a limited time of early development, termed neural crest cells. These cells form while the early structures which will come to form the nervous system grow. They migrate to the front of the embryo to form the face. The cells of patients bearing a mutation of *ALX1* were found to be more likely to die when compared with cells derived from healthy donors. They were also found to show a migration defect. Similar differences were observed in the zebrafish models of the disease created by a disruption of the same gene.

#### Impact
Understanding the causes of malformations of the face will give us the tools to innovate and transform the insufficient treatment options currently available to patients.

and a 405 nm laser at 25% power. Embryos were then placed back into a 28.5°C incubator. Both the photoconverted red (488 nm) and nonphotoconverted green (572 nm) neural crest populations were captured at 4 days post-fertilization (dpf) using a Leica Sp8 and analyzed with the Leica Application Suite X (Leica Microsystems, Buffalo Grove, IL) software for image capturing. A composite image was subsequently generated using ImageJ (NIH; Fig 8D, Film 3).

### Statistical analysis

Each experiment was performed on six independent healthy control $ALX1^{165L/165L}$ clones, three heterozygous $ALX1^{165L/165F}$ clones, and nine homozygous $ALX1^{165F/165F}$ clones, and repeated at least three times. The qualitative craniofacial analysis of $alx1^{-/-}$ zebrafish and Alx1DN injections was performed three times, on three different clutches of embryos. For RT–qPCR experiments, data from each clone were pooled and the mathematical mean was calculated. SEM was used to determine the standard error. To test statistical significance, Student's *t*-test for paired data was used. Statistical analysis of the significance of the qPCR results was performed with an ANOVA test. A *P*-value < 0.05 was considered to be statistically significant. Statistical analysis was performed using GraphPad Prism 7.0 software. The D'Agostino and Pearson normality test was performed to verify normality. For groups that fulfilled normality and equal variance requirements, a one-way ANOVA with a Sidak comparison test (95% confidence interval to compare all the different groups) was performed. For data sets that did not fulfill normality and equal

variance requirements, a Kruskal–Wallis test was performed. Mean values for each group were compared using two-tailed Student's test for comparisons of two independent groups.

## Data availability

The raw sequencing data sets produced in this study are available in the following database: FaceBase Record ID: 25J0 Accession: FB00000907 (https://www.facebase.org/).

Expanded View for this article is available online.

## Acknowledgements

We are grateful for Shriners Hospital for Children (85112) and National Institute of Health (U01DE024443) for funding that supported this work. ECL is a recipient of the Massachusetts General Laurie and Mason Tenaglia Research Scholar award. JK is a recipient of the Shiners Hospital Research Fellowship (84701-BOS-19). We thank Jessica Bathoney for excellent management of our aquatics facility.

## Author contributions

JP, JK, and ECL conceived of the project and designed the research studies. JP, JK, YDH, CT, KK, PY, BY, NC, RM, JC, and YG conducted the experiments. JP, JK, CT, NC, RM, JC, YG, and ECL prepared the manuscript. JK, CT, and ECL worked on the revision of the manuscript.

## Conflict of interest

The author declares that he has no conflict of interest.

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
