## [Review Process File · EMBO Molecular Medicine]

ALX1-related Frontonasal Dysplasia Results From Defective Neural Crest Development and Migration

Jonathan Pini, Janina Kueper, David Hu, Kenta Kawasaki, Pan Yeung, Casey Tsimbal, Baul Yoon, Nikkola Carmichael, Richard Maas, Justin Cotney, Yevgenya Grinblat, and Eric Liao

DOI: [10.15252/emmm.202012013](https://doi.org/10.15252/emmm.202012013)

Corresponding authors: Eric Liao (cliao@partners.org)

Review Timeline:	Submission Date:	10th Jan 20
	Editorial Decision:	29th Jan 20
	Authors' Correspondence:	30th Jan 20
	Editorial Correspondence:	31st Jan 20
	Revision Received:	19th Jun 20
	Editorial Decision:	21st Jul 20
	Revision Received:	12th Aug 20
	Accepted:	13th Aug 20

Editor: Celine Carret

Transaction Report:

29th Jan 2020

Dear Dr. Liao,

Thank you for the submission of your research manuscript to our editorial office. We have now heard back from the referees whom we asked to evaluate your manuscript.

Unfortunately, you will see that both reviewers have very similar and overlapping concerns and while considering the results potentially interesting, raise serious issues regarding the conclusiveness of the data and pinpoint several technical and conceptual problems too that preclude a solid interpretation of the experimental evidence provided. The reviewers call for a considerable amount of additional experimentation to resolve these issues. Therefore, we feel that it would be counter-productive, at this stage to invite a revision of your work as reaching publication level would likely be unachievable in the 3 months-deadline we usually expect.

Given the potential interest of the findings however, we would have no objection to consider a new manuscript on the same topic if at some time in the near future you obtained data that would considerably strengthen the message of the study and address the referees concerns in full. To be completely clear, however, I would like to stress that if you were to send a new manuscript this would be treated as a new submission rather than a revision and would be reviewed afresh (with maybe different referees), in particular with respect to the literature and the novelty of your findings at the time of resubmission. If you decide to follow this route, please make sure you nevertheless upload a letter of response to the referees' comments.

At this stage though, I am sorry to have to disappoint you. I nevertheless hope, that the referee comments will be helpful in your continued work in this area and I thank you for considering EMBO Molecular Medicine.

Yours sincerely,

Celine Carret

Celine Carret, PhD
Senior Editor
EMBO Molecular Medicine

***** Reviewer's comments *****

Referee #1 (Comments on Novelty/Model System for Author):

This is interesting work exploring the use of iPSC-derived NCC to learn about the functions of ALX1 in craniofacial morphogenesis. There is a large volume of work presented and some of it is excellent. The greatest problem with this work is that having obtained their cell lines the authors do not undertake a thorough epigenetic characterisation of them. Rather, they sample just a small number of candidate expression differences and use this as a starting point to derive an excessively fitted model of mechanism.

Referee #1 (Remarks for Author):

This manuscript comprises four parts:

1. Identification of a family segregating a rare homozygous missense variant in ALX1 associated with severe frontonasal dysplasia.
2. Development and differentiation of cells from these individuals with neural crest properties (NCC) via iPSC
3. Exploration of the consequences of knockdown of the orthologous Alx1 gene in zebrafish
4. Functional analysis of migratory properties of the NCC using a scratch assay and the effects of adding bone morphogenetic proteins (BMPs)

The first three parts of the work are broadly convincing, although some of the data should be better presented. The human genetics work is very cursory in its presentation with very little context being provided for the missense variant identified. The causative nucleotide substitution is incorrectly annotated (should be c.493C>T) and corresponding gene accession number is never mentioned, nor is the fact that it is absent in large databases of genomic variation such as gnomAD. As a minimum, I would wish to know the ethnic background of the family and the pedigree to show the results of genotyping each individual. Much more care is needed to place the L165F variant in its proper context. For example, how does it relate to other pathogenic variants previously identified in this gene? Whereabouts in the homeodomain does the Leu residue reside and what are the predicted consequences of substitution there? Are there precedents for pathogenic Leu to Phe substitutions occurring in other homeodomain-containing proteins? The authors state at the end of this section it is "predicted to cause loss of function", but present no evidence for this (indirect or direct) whatever. What would the mechanisms for this be - protein instability or something more specific? In summary, how can they be sure that the homozygous substitution creates a full knockout rather than a hypomorphic allele? In the absence of evidence it would be scientifically more accurate (and safer) to refer to this allele as (for example) ALX1165F/165F (as in legend to Fig.1), rather than ALX1-/-

The description of the iPSC/NCC derivation seemed clear and results were based on independent analyses of 3 separately picked clones for each genotype. Can the authors clarify whether each line was checked by SNP array to exclude major acquired copy number changes? They only mention resequencing of ALX1.

The authors demonstrate that the mutant NCC lines could differentiate to all three mesenchymal lineages (adipose, bone, cartilage), yet there also appeared to be evidence of an early differentiation block. Presumably block this could be overcome by culturing in specific conditions known to yield differentiated cell types. To explore the differences between the WT and ALX1165F/165F lines further the authors used real-time RT-PCR of candidate genes, but they do not appear to have used RNA-seq (or any other genome-wide measure such as ATAC-seq) to obtain an overall, unbiased picture of differences in expression or epigenetic landscape. This naturally leads them to explore only a very limited range of possibilities to explain the functional differences. In my view this is an important missed opportunity to gain a real and deep understanding of the epigenetic differences between the cells.

In the zebrafish, the authors made two knockout alleles using CRISPR/cas9 and surprisingly found these to be viable and mostly normal, but with a low prevalence of head defects. (Previous data (Dee et al 2013) related to use of morpholinos, which the authors correctly state to be a less clean system for genetic perturbation.) They showed upregulation of paralogous Alx transcripts suggesting functional redundancy. The authors then introduced a truncated Alx1 copy which appeared to have dominant negative activity. Additionally, by measuring pax3a and pax3b

expression in mutants and examining the effect of embryo injection of pax3a and pax3b, the authors aim to strengthen the functional link between ALX1 and PAX3. However, the evidence remains indirect and it isn't clear whether levels of pax3 mRNA used was physiological. The experimental narrative jumps around between NCC and zebrafish model (both Figs 3 and 4 show a mixture of NC/zebrafish data), which made the line of argument hard to follow in some places. The purported difference in a migration defect shown in Fig 4d as not fully obvious to this reviewer from the single example image shown in Fig. 4d, nor is it obvious that this arises from abnormal migration rather than disturbed cell division.

Returning to the NCC lines, the authors use a scratch assay as a functional readout and propose that PAX3 is repressed by ALX1 and that PAX3 overexpression phenotypes the effect of ALX1 knockdown. Furthermore they identify a decrease of BMP2 and excess of BMP9 in mutant media, and show that respective supplementation and antagonism partially rescues the functional deficiency. The work is summarised with a "wiring diagram" (Fig 6). In this last part of the work, I found the authors too keen to create a neat "story" from the work, this involved them cutting many corners. Alx1, as might be expected of a homeodomain transcription factor, has a very large number of transcriptional targets. For example the authors don't quote recent work (Khor et al Development 2019) investigating this further (this work was done of sea urchins, but the principle that Alx1 likely has thousands of targets in mammals holds true). Regarding the identification of PAX3 as a specific target, (the introduction states: "we determined that ALX1 is a transcriptional repressor of PAX3", NO data are presented to support the direct effect implied. The transfection experiments performed on NCC lines represent a highly unphysiological system and the authors don't attempt to measure whether the change in PAX3 expression achieved matched physiological expectations. It was also unclear how the authors specifically identified imbalances in BMP2 and BMP9 (what about all the other BMPs?), and why these would have antagonistic effects is not clearly discussed. Given that this work originated from a lab in an English-speaking country, there are an extraordinarily high number of grammatical errors - far too many to point out individually- suggesting that due diligence has not been shown by the senior author in reviewing this work before submission for publication.

Specific points: when relating to the human gene, ALX1 should be italicised

All abbreviations should be defined on first use (there is even one in the title!)

Citations to authors in text sometimes include years and sometimes do not

Introduction - it is implied that frontoasal dysplasia is necessarily associated with facial clefting, which is not always the case.

Ephrin is not a "receptor tyrosine kinase"

Methods

What is "26omega26en"?

Correct gene name for "36B4" is RPLP0

Referee #2 (Comments on Novelty/Model System for Author):

More quantification is needed.

Referee #2 (Remarks for Author):

This manuscript by Pini et al investigates the role for ALX1 in Human iPSCs and zebrafish craniofacial development and frontonasal dysplasia (FND). Starting from patient samples, iPSCs are derived from peripheral blood from control and affected patients. The authors show that the cells

can be differentiated into neural crest cells (NCC), and their derivatives. Examination of expression of NCC genes show an increase in early NCC markers, including PAX3 and ZIC1. Generation of a zebrafish mutant allele and dominant negative constructs results in mild and strong craniofacial phenotype respectively, which the authors argue is due to a change in gene expression and migration via a switch in BMP signaling. The manuscript presents an interesting approach starting from patients to animal models. However, there are some issues in the interpretation and presentation that preclude publication at this time. My comments are below.

1. The writing is awkward in many places with typos and acronyms used without definition. The first time these are used in the text, they should be defined. If not, it difficult for the reader. In addition, there is no to little discussion of the zebrafish experiments in the abstract and introduction. Also, there are some overstatements that should be toned down, including "largest pedigree", "Alx functions as a dominant negative" since you are using this construct to remove Alx function (?), "FN defect" which I actually don't know what that means. Also the title should reflex the whole body of results found in the manuscript. Finally, the discussion is a summary of results and does not place this work in the context of the larger field. This should be shortened and focus on what the result mean.

2. Because ALX genes are known in Human and animal models to play a role in FND, what is presented is not completely novel. However, I think this is a novel mutation, but this is not discussed. What is known about the existing Human mutations in ALX1 that are causative for FND and are the mutations in the same position as this family? What happens in iPSCs of the known mutations?

2. Some additional quantification should be considered to increase the rigor of the work. Besides the qRT-PCR (needed in S5b however) much of the description of the phenotypes are not quantified ie: Cell shape of iPSCs, zebrafish alcian blue cartilage measurements, zebrafish migration, etc.

3. The use of iPSCs is a unique addition, however there is little description of how the days in culture correlate to developmental age. Do you observe a progression of gene expression as in vivo, ie neural plate border, neural crest specifiers, epithelial to mesenchymal transition, migration, and differentiation in this time frame? If so, the fact that CD57 stays on does not affect the ability of differentiation. And the expression of pax3a does not fit this model whereas pax3b does. However, the pan overexpression of pax3a is more severe then pax3b. What is the explanation for this? Also, there should be a discussion about the use of non-tissue or temporal specific RNA and what the caveats are.

4. The rationale for choosing pax3 as a focus is not clear, since some of the other genes are more significantly changed. In addition, the best experiment to test genetic epistasis is to create a double knockout with Alx1 and Pax3. This would determine if they genetically interact and is the most definitive proof. This should be considered.

5. It seems to me that the focus should be on cell death and not migration. What cells were transfected in Figure 4? What do the zebrafish look like when assayed for cell death? Only one experiment is shown in Figure 4a. The fact that there are less cells overall and/or they are dying would complicate the analysis on migration. In the zebrafish, it does seem that the cells are there, but not able to coalesce in the correct location. Are they dying? Similarly, the focus on BMP as signaling factor to focus on is not clear from the text as it seems to come out of nowhere. While of course BMP signaling is important for NCC specification, the relevance of the switch from BMP2 to

BMP9 is not clear. Treatment of the cells helps but does not seem to completely close the cell wound. What does treatment of wildtype cells with these same doses do to the cells? Is the cell death rescued in the *Alx1*^{-/-} cells?

Overall, there are many intriguing results, however, answers to all these questions are required before publication.

Dear Celine,

Thank you very much for the kind message. I have reviewed the comments and the critiques are not as problematic as perhaps the tone of reviewer #1, probably because of the subpar writing, of which I take full responsibility and will address thoroughly. The good news is that we have already done the RNA-seq in human iPSC, and a collaborator has done single cell RNA-seq in zebrafish, and the intersection of this data supports our work. The other assays can all be done in the lab as we have the iPSC cultures ready to go in our ongoing work. After an itemized review of the critiques, I do believe we can revise the manuscript within 3 months.

Would you consider returning this work to the same reviewers in a MAJOR REVISION rather than a new manuscript review that will potentially have new reviewers and fresh new comments? I would also like the reviewers to see that we have taken their comments to heart and completed a major revision, and the project is improved from their suggestions. No matter what we are committed to improving this work, and would prefer your consideration for a major review.

Thank you very much for your consideration.

All the best,

Eric Liao

31st Jan 2020

Dear Dr. Liao,

Thank you for your kind email asking us to reconsider our decision.

Given that you already have done a lot of work on the study and also considering your arguments, we are happy to allow the manuscript to get back in the system as a major revision. Don't worry about the 3-months deadline, this is a wish but not a must, it will be fine for us if it takes a little longer. If however you anticipated a delay over 6-months, I'd ask that you get in touch with us as our scooping protection may therefore no longer apply.

I wish you all the best with the revision and thank you for contacting us.

With my best wishes,
Celine

Celine Carret, PhD
Senior Editor
EMBO Molecular Medicine

ITEMIZED RESPONSES

EMM-2020-12013-V2-Q

Reviewer 1

We appreciate the reviewer's comments. The following are our point-by-point responses:

- 1) *The greatest problem with this work is that having obtained their cell lines the authors do not undertake a thorough epigenetic characterisation of them.*

RESPONSE: It has been reported that the epigenetic landscape is reset during the generation of induced pluripotent stem cells (iPSC)(Liang & Zhang, 2013). Variations in the process, as well as prolonged culture time, may nonetheless result in epigenetic differences between iPSC lines. Since the reprogramming process is the most time- and cost consuming component of iPSC model generation, repeating the reprogramming itself was cost-prohibitive. Epigenetic differences were controlled for as best as possible by ensuring the following:

- All major experiments were performed in a minimum of three clones
- All major experiments were repeated at least three times
- All clones used for experiments were at the same passage number
- None of the clones used for experiments exceeded passage cycle 15

These points are now recorded in the 'Methods' section (line 510). On a separate note, variable X-chromosome inactivation can result in differences between female iPSC lines. To increase the number of affected subjects enrolled in the study, the choice was made not to exclude the female affected subject in order to allow us to verify findings between the two affected family members for whom iPSC lines were generated.

- 2) *[...] they sample just a small number of candidate expression differences and use this as a starting point to derive an excessively fitted model of mechanism.*

RESPONSE: We agree that a major shortcoming of the original manuscript was the fitted mechanistic model based on published literature. To emphasize the findings in the Neural Crest Cells (NCC) which form the most substantial results of the experiments performed for this project, the hypothesis that *PAX3* is regulated by *ALX1* and the preliminary data indicative of this relationship was removed from the manuscript. The experiments required to adequately address the role of *ALX1* within the human NCC gene regulatory network, namely the generation of an isogenic line that would allow for well-controlled RNA-seq and ChIP-seq or CUT&RUN experiments for an unbiased analysis were beyond the scope of this

study, as our laboratory closed for the last 3 months due to the pandemic. The transcriptional analysis of candidate genes indicative of particular stages of NCC differentiation was retained in order to survey the developmental progression created *in vitro* with iPSC-derived NCC. The importance of this data point has been reworded, with the lack of transcriptional data from human embryos to compare our *in vitro* data with being critically discussed in the new discussion section 'iPSC for craniofacial disease modeling' (line number 413).

3) *The human genetics work is very cursory in its presentation with very little context being provided for the missense variant identified.*

RESPONSE: A diagram of the *ALX1* gene and protein are now included in a revised Figure 1, which gives a visual overview of the missense variant of this manuscript's pedigree in context with other variants reported in the literature. Sections on the *in silico* prediction of the pathogenicity of the missense variant via a number of different algorithms, the placement of the variant within the protein, and the likely consequences of the amino acid substitution are now included in both the 'Results' and the 'Discussion' sections. The case reports of other families affected by variants or deletion of *ALX1* are now discussed in depth in the 'Discussion' section (line 349).

4) *The causative nucleotide substitution is incorrectly annotated (should be c.493C>T) and corresponding gene accession number is never mentioned, nor is the fact that it is absent in large databases of genomic variation such as gnomAD.*

RESPONSE: Both the annotation and the placement of the nucleotide substitution were noted to be incorrect (c.493C>T) and corrected to c.648C>T throughout the manuscript. The accession number of *ALX1* as well as other genes related to Frontonasal Dysplasia (FND) and their respective disorders in the format of the Online Mendelian Inheritance in Man (OMIM) database are now included in the 'Introduction' section. The novel missense variant identified in this study has been researched on gnomAD, with the findings reported and discussed in both the 'Results' and the 'Discussion' sections.

- *As a minimum, I would wish to know the ethnic background of the family and the pedigree to show the results of genotyping each individual*

RESPONSE: The family's ethnic background is Amish, which is now mentioned in the manuscript alongside the note that the parents are blood relatives (first cousins). The pedigree has been annotated accordingly in the revised version of Figure 1. The information on which subjects were genotyped, as well as their phenotype, is now addressed in greater

detail in the 'Results' section. To make the information more accessible to the reader, Figure 1 was further modified through a numbering of the subjects enrolled in this study to allow for easier referencing of the individuals and their respective phenotypes and genotypes.

- *Much more care is needed to place the L165F variant in its proper context. For example, how does it relate to other pathogenic variants previously identified in this gene?*

RESPONSE: Extensive discussion of the two case studies which describes a total of three families affected by three different *ALX1* variants has now been included in the beginning of the 'Discussion' section (line number 349). These variants have been included in the revised version of Figure 1 which gives a visual overview of the missense variant of the pedigree reported on in this manuscript in the context of the variants already reported in the literature.

- *Whereabouts in the homeodomain does the Leu residue reside and what are the predicted consequences of substitution there?*

RESPONSE: As can be seen in the gene diagram, the Leu residue resides centrally within the homeobox domain on helix II. No data on the crystallographic ultrastructure of the *ALX1* protein itself has been published to date. To assess the predicted consequences of the substitution caused by the L165F variant identified in the pedigree, a number of bioinformatic tools were used to assess the pathogenicity of the missense mutation (Sift, Polyphen, muttaster, fathmm). This information is now included in the appropriate 'Results' and 'Methods' sections. The location of the L165F likely disrupts the ability of the protein to regulate the transcription of downstream targets of this transcription factor. This information is now included and contrasted with another case report which identifies a missense variant in helix III in another family in the 'Discussion' section.

- *Are there precedents for pathogenic Leu to Phe substitutions occurring in other homeodomain-containing proteins?*

RESPONSE: Leucine is an aliphatic, branched amino acid while phenylalanine is an aromatic, neutral and nonpolar amino acid. Due to properties of leucine, the substitution itself is likely disruptive to helix II in the DNA-binding element of the homeodomain within which it resides. Disruptive leucine to phenylalanine substitutions have been described in a number of published, genotyped disorders. Independently of this, missense variants within the homeodomains of both *ALX3* and *ALX4* have been connected to FND. All of the information above is now recorded in the 'Discussion' section (line 349).

5) *The authors state at the end of this section it is "predicted to cause loss of function", but present no evidence for this (indirect or direct) whatever. What would the mechanisms for this be - protein instability or something more specific? In summary, how can they be sure that the homozygous substitution creates a full knockout rather than a hypomorphic allele? In the absence of evidence it would be scientifically more accurate (and safer) to refer to this allele as (for example) ALX1^{165F/165F} (as in legend to Fig.1), rather than ALX1^{-/-}.*

RESPONSE: As mentioned above, we could not identify preexisting data on the crystallographic ultrastructure of the ALX1 protein itself. In consequence, we performed a bioinformatics analysis that focus on the algorithmic assessment of our missense variant in the context of conservation. To assess the predicted consequences of the substitution caused by the L165F variant identified in the pedigree, a number of bioinformatic tools were used to assess the pathogenicity of the missense mutation (Sift, Polyphen, muttaster, fathmm). This information is now included in the appropriate 'Results' and 'Methods' sections. The L165F variant lies within helix II of the DNA binding domain within the homeobox of ALX1, likely disrupting the ability of the protein to regulate the transcription of downstream targets of this transcription factor. This information is now included and contrasted with another case report which identifies a missense variant in helix III in another family in the 'Discussion' section. When performing qPCR on our patient's cell lines and compared them to their parents as well as healthy control lines, we found significant differences in the levels of *ALX1* transcripts, as seen in Figure 3, We agree with the reviewer that this data points more towards a hypomorphic allele (given the fact that there is no absence of *ALX1* transcripts in the affected subjects) than a total loss-of-function, and have corrected the notation of the affected subject's cells from *ALX1^{-/-}* to *ALX1^{165F/165F}* throughout the manuscript and the figures.

6) *Can the authors clarify whether each line was checked by SNP array to exclude major acquired copy number changes?*

RESPONSE: We performed genetic stability testing on our cell lines to exclude copy number variations in addition to sequencing *ALX1* in order to confirm the retention of the line's original variant.

7) *The authors demonstrate that the mutant NCC lines could differentiate to all three mesenchymal lineages (adipose, bone, cartilage), yet there also appeared to be evidence of an early differentiation block. Presumably this block could be overcome by culturing in specific conditions known to yield differentiated cell types. To explore*

the differences between the WT and ALX1165F/165F lines further the authors used real-time RT-PCR of candidate genes, but they do not appear to have used RNA-seq (or any other genome-wide measure such as ATAC-seq) to obtain an overall, unbiased picture of differences in expression or epigenetic landscape. This naturally leads them to explore only a very limited range of possibilities to explain the functional differences. In my view this is an important missed opportunity to gain a real and deep understanding of the epigenetic differences between the cells.

RESPONSE: The experiments required to adequately address the role of *ALX1* within the human NCC gene regulatory network, namely the generation of an isogenic line that would allow for well-controlled RNA-seq and CHIP-seq or CUT&RUN experiments for an unbiased analysis were beyond the scope of this study. The transcriptional analysis of candidate genes indicative of particular stages of NCC differentiation were retained in order to survey the developmental progression created *in vitro* with iPSC-derived NCC. The importance of this data point has been deemphasized, with the lack of transcriptional data from human embryos to compare our *in vitro* data with being critically discussed in the new discussion section 'iPSC for craniofacial disease modeling' (line number 413).

8) *In the zebrafish, the authors made two knockout alleles using CRISPR/cas9 and surprisingly found these to be viable and mostly normal, but with a low prevalence of head defects. (Previous data (Dee et al 2013) related to use of morpholinos, which the authors correctly state to be a less clean system for genetic perturbation.) They showed upregulation of paralogous Alx transcripts suggesting functional redundancy. The authors then introduced a truncated Alx1 copy which appeared to have dominant negative activity. Additionally, by measuring pax3a and pax3b expression in mutants and examining the effect of embryo injection of pax3a and pax3b, the authors aim to strengthen the functional link between ALX1 and PAX3. However, the evidence remains indirect and it isn't clear whether levels of pax3 mRNA used was physiological.*

RESPONSE: In zebrafish, over-expression by mRNA injection does not attempt to dose physiologic level of mRNA. Some toxic side effects from the quantity of mRNA injected are to be expected. Supplemental Figure 5 accounts for this phenomenon, displaying a stratification of embryo death by quantity of injection. The hypothesis that zebrafish *pax3* is regulated by *alx1* and the preliminary data indicative of this relationship was removed from the manuscript. The experiments required to adequately address the role of *alx1* within the NCC gene regulatory network were beyond the scope of this study.

9) *The experimental narrative jumps around between NCC and zebrafish model (both Figs 3 and 4 show a mixture of NC/zebrafish data), which made the line of argument hard to follow in some places.*

RESPONSE: In order to clarify, all zebrafish data is now clearly separated from cell data in both the manuscript and the figures. The results are then discussed in a more integral manner in the Discussion.

10) *The purported difference in a migration defect shown in Fig 4d as not fully obvious to this reviewer from the single example image shown in Fig. 4d, nor is it obvious that this arises from abnormal migration rather than disturbed cell division.*

RESPONSE: To allow for an easier visualization of the population of NCC delayed in their migration following laser conversion, schematics demonstrating the distribution and placement of the delayed cells within the zebrafish's craniofacial region have been added to the new Figure 6 summarizing our findings in zebrafish. Additionally, a new video showing a 3D Z-stack of a zebrafish displaying delayed frontonasal NCC migration following laser conversion has been added to the revised submission (Film 3). To allow red-green blind readers to detect the difference more clearly, the NCC visualized in the new film are displayed in magenta rather than red. These results show a heterotopic location of the NCC in the later timepoint at 4 dpf, which can be explained by altered migration but not altered cell division. Altered cell division may result in fewer NCC in the final destination, but this was not observed. Altered cell division would also not result in the NCC being observed in a new location rather than the median anterior neurocranium location.

11) *Returning to the NCC lines, the authors use a scratch assay as a functional readout and propose that PAX3 is repressed by ALX1 and that PAX3 overexpression phenotypes the effect of ALX1 knockdown. Furthermore they identify a decrease of BMP2 and excess of BMP9 in mutant media, and show that respective supplementation and antagonism partially rescues the functional deficiency. The work is summarised with a "wiring diagram" (Fig 6). In this last part of the work, I found the authors too keen to create a neat "story" from the work, this involved them cutting many corners.*

RESPONSE: We apologize that the original manuscript summarized a lot of data, both from the literature and from our experiments. The wording and sections presenting a fitted narrative has been removed.

12) *Alx1, as might be expected of a homeodomain transcription factor, has a very large number of transcriptional targets. For example the authors don't quote recent work (Khor et al Development 2019) investigating this further (this work was done of sea urchins, but the principle that Alx1 likely has thousands of targets in mammals holds true).*

RESPONSE: The work by Khor et. al explores targets of *ALX1* using a customized antibody for a ChIP-sequencing experiment on mesenchymal precursor cells in the sea urchin. This provides an interesting context for our work, since this study explores the role of *ALX1* as a transcription factor, though the determination of the protein's regulatory targets in the zebrafish or the human cell lines was not within the scope of this project. A discussion of the study was added to the discussion section (line 416).

13) *Regarding the identification of PAX3 as a specific target, (the introduction states: "we determined that ALX1 is a transcriptional repressor of PAX3", NO data are presented to support the direct effect implied.*

RESPONSE: The hypothesis that *PAX3* is regulated by *ALX1* and the preliminary data indicative of this relationship was removed from the manuscript.

14) *The transfection experiments performed on NCC lines represent a highly unphysiological system and the authors don't attempt to measure whether the change in PAX3 expression achieved matched physiological expectations.*

RESPONSE: The hypothesis that *PAX3* is regulated by *ALX1* and the preliminary data indicative of this relationship was removed from the manuscript.

15) *It was also unclear how the authors specifically identified imbalances in BMP2 and BMP9 (what about all the other BMPs?), and why these would have antagonistic effects is not clearly discussed.*

RESPONSE: Language to clarify the methodology regarding the identification of BMP levels was added in the 'Materials and Methods' section (line 704). A large section was added to the 'Discussion' section to describe the current state of knowledge on BMP2 and BMP9 (line 474).

16) *Given that this work originated from a lab in an English-speaking country, there are an extraordinarily high number of grammatical errors - far too many to point out individually- suggesting that due diligence has not been shown by the senior author in reviewing this work before submission for publication.*

RESPONSE: We sincerely apologize for the high number of grammatical errors and lack of sentence structure in our manuscript. We have rewritten the entire manuscript.

17) Specific points: when relating to the human gene, ALX1 should be italicised

RESPONSE: When used in reference to the gene, all mentions of *ALX1* have now been italicized.

18) All abbreviations should be defined on first use (there is even one in the title!).

RESPONSE: All abbreviations are now defined on first use. All abbreviations were removed from the title. Furthermore, the title as well as the short title were edited for clarity. The title is now 'ALX1-related Frontonasal Dysplasia Results From Defective Neural Crest Cell Development and Migration'. The short title is now 'Defective Neural Crest Cells Cause ALX1-related Frontonasal Dysplasia'.

19) Citations to authors in text sometimes include years and sometimes do not

RESPONSE: All references have now been formatted in a standardized manner in accordance with the Journal's policy.

20) Introduction - it is implied that frontonasal dysplasia is necessarily associated with facial clefting, which is not always the case.

RESPONSE: This incorrect statement has been removed, and replaced with a statement that notes facial clefting as one potential component of frontonasal dysplasia (line 74).

21) Ephrin is not a "receptor tyrosine kinase"

RESPONSE: This erroneous statement has been removed (line 80).

22) Methods: What is "26romega26en"?

RESPONSE: The erroneous statements in question have been removed.

23) Correct gene name for "36B4" is RPLP0

RESPONSE: All mentions of "36B4" have been replaced with "RPLP0" throughout the manuscript. It came to our attention that the gene "SNAI2" was erroneously presented as "SLUG". This gene name, too, was corrected throughout the manuscript.

Reviewer 2

We appreciate that the reviewer's comments. The following are our point-by-point responses:

1) *More quantification is needed.*

RESPONSE: All Figures as well as the 'Results' section of the manuscript have been extensively edited to highlight the more quantitative aspects of this work.

2) *The writing is awkward in many places with typos and acronyms used without definition. The first time these are used in the text, they should be defined. If not, it difficult for the reader.*

RESPONSE: We sincerely apologize for the high number of grammatical errors and lack of sentence structure in our manuscript. We have rewritten the entire manuscript. All abbreviations are now defined on first use. All abbreviations were removed from the title.

3) *[...] there is no to little discussion of the zebrafish experiments in the abstract and introduction.*

RESPONSE: A more in-depth discussion of zebrafish experiments is now included in both the abstract and the introduction.

4) *[...] there are some overstatements that should be toned down, including "largest pedigree", "Alx functions as a dominant negative" since you are using this construct to remove Alx function (?), "FN defect" which I actually don't know what that means.*

RESPONSE: During the rewriting of the manuscript, overstatements were removed. Specifically, the statement "largest pedigree" was removed; the statement "Alx functions as a dominant negative" was edited to reflect the hypothesis in question (line 311); the term "FN defect" was removed.

5) *[...] the title should reflex the whole body of results found in the manuscript.*

RESPONSE: The title as well as the short title were edited for clarity. The title is now 'ALX1-related Frontonasal Dysplasia Results From Defective Neural Crest Cell Development and Migration'. The short title is now 'Defective Neural Crest Cells Cause ALX1-related Frontonasal Dysplasia'.

6) [...], the discussion is a summary of results and does not place this work in the context of the larger field. This should be shortened and focus on what the result mean.

RESPONSE: All summaries of results were removed to avoid redundancy. The Discussion was entirely rewritten to reflect points of interesting with respect to the context of the larger field. Specifically, the discussion is now divided into the following subsections:

- Human genetics of ALX1 (line 350)
- The ALX gene family: ALX1, ALX3, and ALX4 (line 399)
- iPSC for craniofacial disease modeling (line 422)
- Animal models of ALX1-related FND (line 508)
- Conclusion (line 530)

7) *Because ALX genes are known in Human and animal models to play a role in FND, what is presented is not completely novel. However, I think this is a novel mutation, but this is not discussed. What is known about the existing Human mutations in ALX1 that are causative for FND and are the mutations in the same position as this family? What happens in iPSCs of the known mutations?*

RESPONSE: The study now emphasizes that the variant described is novel. All other case reports which describe families affected by ALX1-related FND are now included in the 'Discussion', with a detailed description of the variant identified (line 366). One family was found to have a missense variant in the homeodomain one helix downstream from the helix II affected by the L165F variant identified in this study's pedigree. Unfortunately, the creation of gene-edited iPSC lines that recreate the three different variants of ALX1 described in the published literature was not within the scope of this project.

8) *Some additional quantification should be considered to increase the rigor of the work. Besides the qRT-PCR (needed in S5b however) much of the description of the phenotypes are not quantified ie:*

- Cell shape of iPSC,

RESPONSE: Due to the COVID-19 pandemic, we are currently unable to access the microscopes required to visually analyze the shape of the NCC we reference. We have therefore added a sentence emphasizing the qualitative nature of this observation (line 196).

The lineage tracing in the zebrafish model was repeated in order to present superior visualization of the findings at a greater magnification. This data is shown in a new film displaying a 3D Z-stack of a zebrafish displaying delayed frontonasal NCC migration

following laser conversion (Film 3). To allow red-green blind readers to detect the difference more clearly, the NCC visualized in the new film are displayed in magenta rather than red.

9) *The use of iPSCs is a unique addition, however there is little description of how the days in culture correlate to developmental age. Do you observe a progression of gene expression as in vivo, ie neural plate border, neural crest specifiers, epithelial to mesenchymal transition, migration, and differentiation in this time frame?*

RESPONSE: The study includes a survey of NCC marker genes throughout the differentiation protocol. For this, qPCR was performed for the neural plate border specifiers *ZIC1*, *PAX7*, *PAX3*, *MSX1*, *MSX2*, and *DLX5*; the neural crest specifiers *FOXD3*, *P75*, *TFAP2A*, *SNAI2*, and *TWIST1*; the lineage specifier *HAND2*; and *ALX1*. The analysis identified a progression of gene expression that appears to mirror that reported in animal studies. To make this information more accessible to the reader, the section detailing these results was fundamentally reorganized (line 145). A new figure was constructed to show the gene profiles in a more organized manner. A new section in the 'Discussion' section now details the challenges of relating NCC derived *in vitro* with *in vivo* data (line 432).

10) *If so, the fact that CD57 stays on does not affect the ability of differentiation.*

RESPONSE: It is correct that the ability of NCC to differentiate into a number of mesenchymal lineages was unchanged between different cell lines, with no significant differences found. This may be explained in two possible ways: either the pathways affected by this study's *ALX1* variant are not implicated in mesenchymal differentiation, or, perhaps more likely, the directed differentiation that occurs with the chemically defined mediums for *in vitro* differentiation allow for a circumnavigation of the pathways that normal *in vivo* development wouldn't allow for.

11) *And the expression of pax3a does not fit this model whereas pax3b does. However, the pan overexpression of pax3a is more severe than pax3b. What is the explanation for this?*

RESPONSE: The hypothesis that *PAX3* is regulated by *ALX1* and the preliminary data indicative of this relationship was removed from the manuscript. The experiments required to adequately address the role of *ALX1* within the human NCC gene regulatory network, namely the generation of an isogenic line that would allow for well-controlled RNA-seq and ChIP-seq or CUT&RUN experiments for an unbiased analysis, were beyond the scope of this study.

12) *Also, there should be a discussion about the use of non-tissue or temporal specific RNA and what the caveats are.*

RESPONSE: The hypothesis that *PAX3* is regulated by *ALX1* and the preliminary data indicative of this relationship was removed from the manuscript.

13) *The rationale for choosing pax3 as a focus is not clear, since some of the other genes are more significantly changed. In addition, the best experiment to test genetic epistasis is to create a double knockout with Alx1 and Pax3. This would determine if they genetically interact and is the most definitive proof. This should be considered.*

RESPONSE: The hypothesis that *PAX3* is regulated by *ALX1* and the preliminary data indicative of this relationship was removed from the manuscript.

14) *What cells were transfected in Figure 4?*

RESPONSE: The hypothesis that *PAX3* is regulated by *ALX1* and the preliminary data indicative of this relationship was removed from the manuscript.

15) *The fact that there are less cells overall and/or they are dying would complicate the analysis on migration. In the zebrafish, it does seem that the cells are there, but not able to coalesce in the correct location. Are they dying?*

RESPONSE: The NCC location in a heterotopic region outside of the anterior neurocranium indicates a fundamental problem in cell migration. Whether additional cellular processes such as cell death or decreased cell division can also be operating cannot be ruled out. However, cell death or decreased cell division will not result in the NCC migrating to an altered position, instead culminating in fewer cells reaching the expected location. This explanation has been added to the manuscript.

16) *Similarly, the focus on BMP as signaling factor to focus on is not clear from the text as it seems to come out of nowhere. While of course BMP signaling is important for NCC specification, the relevance of the switch from BMP2 to BMP9 is not clear. Treatment of the cells helps but does not seem to completely close the cell wound.*

RESPONSE: It is correct that the treatment of the cells did not completely rescue the migration defect. We believe that a pretreatment of the NCC over several hours (rather than a supplementation with BMP2 or CV2 at the beginning of the migration experiment) could potentially result in a complete rescue of the migration defect. A new section extensively discussing the rationale of BMP signaling in the *in vitro* NCC model of FND has been added to the discussion (line 475).

17) *What does treatment of wildtype cells with these same doses do to the cells?*

RESPONSE: Due to the COVID-19 crisis, we were unfortunately unable to treat wildtype cells with the same doses of BMP2 and CV2 as our initial set of experiments. That said, other experiments on wildtype cells with soluble BMP2 and CV2 published in the literature have used higher concentrations than we did in our experiments, without reporting toxic effects (Gao, Cheng et al., 2019, Yao, Jumabay et al., 2012).

18) *Is the cell death rescued in the *Alx1*^{-/-} cells?*

RESPONSE: Due to the COVID-19 crisis, we were unfortunately unable to perform FACS analysis to examine the rate of apoptosis in subject-derived NCC treated with BMP2 or CV2.

References

- Gao X, Cheng H, Awada H, Tang Y, Amra S, Lu A, Sun X, Lv G, Huard C, Wang B, Bi X, Wang Y, Huard J (2019) A comparison of BMP2 delivery by coacervate and gene therapy for promoting human muscle-derived stem cell-mediated articular cartilage repair. *Stem Cell Res Ther* 10: 346
- Liang G, Zhang Y (2013) Genetic and epigenetic variations in iPSCs: potential causes and implications for application. *Cell Stem Cell* 13: 149-59
- Yao Y, Jumabay M, Ly A, Radparvar M, Wang AH, Abdmaulen R, Bostrom KI (2012) Crossveinless 2 regulates bone morphogenetic protein 9 in human and mouse vascular endothelium. *Blood* 119: 5037-47

21st Jul 2020

Dear Mr. Liao,

Thank you for the submission of your revised manuscript to EMBO Molecular Medicine. We have now received the enclosed reports from the referee who was asked to re-assess it. As you will see the reviewer is now globally supportive and I am pleased to inform you that we will be able to accept your manuscript pending the following final amendments:

1) Please address the minor text change commented by referee 1.

Please address the referee's comments in writing. At this stage, we'd like you to discuss referee's 1 points and if you do have data at hand, we'd be happy for you to include it, however we will not ask you to provide any additional experiments at this stage.

Please provide a point-by-point letter INCLUDING my comments as well as the reviewer's reports and your detailed responses to their comments (as Word file).

Please submit your revised manuscript within two weeks.

I look forward to reading a new revised version of your manuscript as soon as possible.

Yours sincerely,

Celine Carret

Celine Carret, PhD
Senior Editor
EMBO Molecular Medicine

***** Reviewer's comments *****

Referee #1 (Comments on Novelty/Model System for Author):

1. Ms is much improved in technical quality as a result of the re-write and removal of poorly supported data
2. First derivation of NCC to study cellular mechanisms of ALX1-related malformation. Functional insights into mechanism with identification of effects of BMPs on correcting defective cell migration phenotype. Novel work on zebrafish mutants.
3. No medical impact in short-medium term, except to establish the correct genetic diagnosis in one family based on known disease gene. Therapeutic use distant because the severe congenital malformations develop during early pregnancy.
4. Combination of human NCC and zebrafish good, complementary model systems

Referee #1 (Remarks for Author):

The authors have taken seriously the comments and criticisms of two reviewers and accordingly have extensively revised the manuscript. The result is a much improved version that is easier to read and follow and is more realistic in its conclusions, highlighting limitations of the study.

The authors should address the following points.

L29 and elsewhere: to accordance with recommended HGVS nomenclature

(<http://varnomen.hgvs.org/>), authors should refer to the mutant protein as p.L165F.

L110, 636: Correct notation of the cDNA is indeed c.493C>T as per previous review. This is because cDNA numbering starts at the ATG initiator codon (see HGVS nomenclature). Reference to Fig 1C will show that the numbering used by authors would place their variant after the c.531+1G>A mutation, whereas it is correctly placed before it. The Genbank accession of human ALX1 (still not provided) is NM_006982.3.

L118: "with haploinsufficiency consistent with the observed autosomal recessive inheritance pattern": This statement is by definition incorrect, as haploinsufficiency describes the situation in which heterozygous loss-of-function manifests with a phenotype. Since the carrier parents do not manifest any phenotype, this is not a haploinsufficiency.

L383 and L412: The references provided to document the contributions of ALX4 mutation to FND are not appropriate for this purpose (these references describe parietal foramina). References to ALX4-related FND can be found in the corresponding OMIM entry.

L393: the donor splice site mutation is incorrectly written (should be c.531+1G>A)

L427: reference to Khor et al has been added, but is not included in the references

L575: Please indicate method used to analyse copy number variants

L761: what was the sequence of the targeting sgRNA?

Other minor comments: references still lack year of publication in some places (eg L47)

Numbers and units of measurement should be separated by spaces (occurs in many places, eg L268, 316, 598 and others)

Lack of subscripts or superscripts in chemical formulas (line 611, 661/2 and others)

Although the English is markedly improved, I noted a number of grammatical errors/typos whilst

going through the ms, for example lines 115 (nor), 154 (patterns), 193 (deleted were investigated), 323 (delete ,), 332 (injection), 391, 474 and 475 (no apostrophe), 686 (manufacturer's), 709 (of), 710 (processed), 736/9,741/2 (a plate shaker), 765 (F0), 817 (experiments)

L759 *alx1* should be italicised

ITEMIZED RESPONSES

EMM-2020-12013-V4

Referee 1

We appreciate the Referee's comments. The following are our point-by-point responses:

- 1) *The authors should address the following points. L29 and elsewhere: to accordance with recommended HGVS nomenclature (<http://varnomen.hgvs.org/>), authors should refer to the mutant protein as p.L165F.*

RESPONSE: The mutant protein has been relabeled p.L165F throughout the manuscript.

- 2) *L110, 636: Correct notation of the cDNA is indeed c.493C>T as per previous review. This is because cDNA numbering starts at the ATG initiator codon (see HGVS nomenclature). Reference to Fig 1C will show that the numbering used by authors would place their variant after the c.531+1G>A mutation, whereas it is correctly placed before it. The Genbank accession of human ALX1 (still not provided) is NM_006982.3.*

RESPONSE: The notation of the cDNA has been corrected. The Genbank accession number is now provided.

- 3) *L118: "with haploinsufficiency consistent with the observed autosomal recessive inheritance pattern": This statement is by definition incorrect, as haploinsufficiency describes the situation in which heterozygous loss-of-function manifests with a phenotype. Since the carrier parents do not manifest any phenotype, this is not a haploinsufficiency.*

RESPONSE: The erroneous mention of haploinsufficiency was removed.

- 4) *L383 and L412: The references provided to document the contributions of ALX4 mutation to FND are not appropriate for this purpose (these references describe parietal foramina). References to ALX4-related FND can be found in the corresponding OMIM entry.*

RESPONSE: The incorrect references have been removed, and replaced with Kayserili et al., 2009.

- 5) *L393: the donor splice site mutation is incorrectly written (should be c.531+1G>A)*

RESPONSE: The donor splice site mutation notation has been corrected.

6) *L427: reference to Khor et al has been added, but is not included in the references*

RESPONSE: The accidental oversight has been corrected.

7) *L575: Please indicate method used to analyse copy number variants*

RESPONSE: This information (genome-wide microarray analysis) has been added to the M&M section.

8) *L761: what was the sequence of the targeting sgRNA?*

RESPONSE: The sequence of the targeting sgRNA is GGAGAGCAGCCTGCACGCGA. This information has been added to the manuscript.

9) *Other minor comments: references still lack year of publication in some places (eg L47)*

RESPONSE: This reference error in our Endnote file has been corrected.

10) *Numbers and units of measurement should be separated by spaces (occurs in many places, eg L268, 316, 598 and others)*

RESPONSE: Numbers and units of measurements are now separated by spaces throughout the manuscript and in the tables.

11) *Lack of subscripts or superscripts in chemical formulas (line 611, 661/2 and others)*

RESPONSE: Lack of subscript and superscript has been corrected.

12) *Although the English is markedly improved, I noted a number of grammatical errors/typos whilst going through the ms, for example lines 115 (nor), 154 (patterns), 193 (deleted were investigated), 323 (delete ,), 332 (injection), 391, 474 and 475 (no apostrophe), 686 (manufacturer's), 709 (of), 710 (processed), 736/9,741/2 (a plate shaker), 765 (F0), 817 (experiments)*

RESPONSE: Thank you very much for your detailed review, these grammatical errors have been corrected.

13) *L759 alx1 should be italicised*

RESPONSE: The zebrafish gene *alx1* has been italicized.

13th Aug 2020

Dear Dr. Liao,

Thank you very much for working with us on the last pending items. We are happy to inform you that your manuscript is accepted for publication and will be sent to our publisher to be included in the next available issue of EMBO Molecular Medicine.

Please read below for additional IMPORTANT information regarding your article, its publication and the production process.

Congratulations on your interesting work,

Celine Carret

Celine Carret, PhD
Senior Editor
EMBO Molecular Medicine

Follow us on Twitter @EmboMolMed
Sign up for eTOCs at embopress.org/alertsfeeds

Corresponding Author Name: Eric C. Liao
Journal Submitted to: EMBO Molecular Medicine
Manuscript Number: EMM-2020-12013-T